# Learning from Natural Language Feedback

**Angelica Chen**[*]                                      *angelica.chen@nyu.edu*
*New York University*

**Jérémy Scheurer**[*]                                    *jeremy@apolloresearch.ai*
*Apollo Research*[†]

**Jon Ander Campos**                                      *jonander@cohere.com*
*New York University; HiTZ Center, University of the Basque Country UPV/EHU*

**Tomasz Korbak**                                         *tomek@anthropic.com*
*New York University; FAR AI; University of Sussex*

**Jun Shern Chan**                                        *junshern@nyu.edu*
*New York University; FAR AI*

**Samuel R. Bowman**                                      *sb6065@nyu.edu*
*New York University; Anthropic PBC*

**Kyunghyun Cho**                                         *kyunghyun.cho@nyu.edu*
*New York University; Genentech; CIFAR LMB*

**Ethan Perez**                                           *perez@nyu.edu*
*New York University; FAR AI; Anthropic PBC*

**Reviewed on OpenReview:** *https://openreview.net/forum?id=xo3hI5MwvU*

## Abstract

The potential for pre-trained large language models (LLMs) to use natural language feedback at inference time has been an exciting recent development. We build upon this observation by formalizing an algorithm for learning from natural language feedback at training time instead, which we call Imitation learning from Language Feedback (ILF). ILF requires only a small amount of human-written feedback during training and does not require the same feedback at test time, making it both user-friendly and sample-efficient. We further show that ILF can be seen as a form of minimizing the KL divergence to the target distribution and demonstrate proof-of-concepts on text summarization and program synthesis tasks. For code generation, ILF improves a CODEGEN-MONO 6.1B model's pass@1 rate from 22% to 36% on the MBPP benchmark, outperforming both fine-tuning on MBPP and on human-written repaired programs. For summarization, we show that ILF can be combined with learning from human preferences to improve a GPT-3 model's summarization performance to be comparable to human quality, outperforming fine-tuning on human-written summaries. Overall, our results suggest that ILF is both more effective and sample-efficient than training exclusively on demonstrations for improving an LLM's performance on a variety of tasks.

---

[*]Equal contribution.
[†]Work done while at NYU.

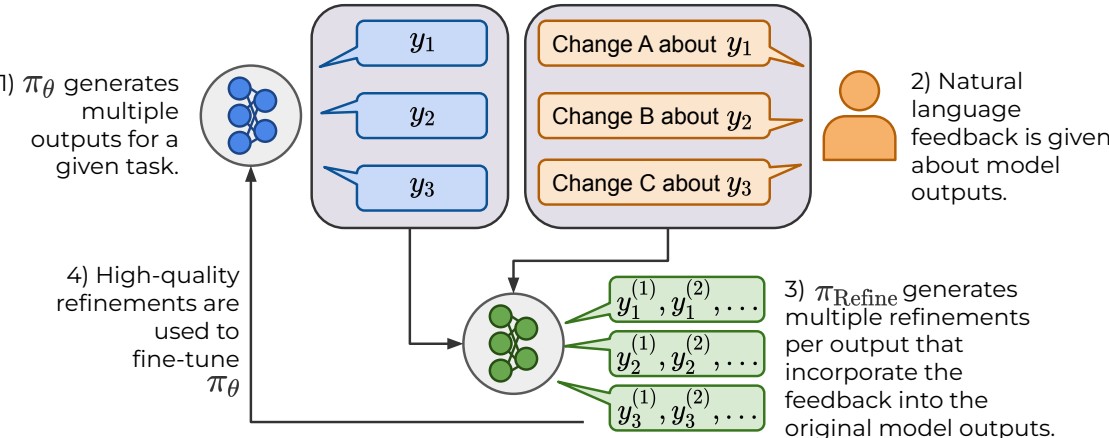

Figure 1: A high-level overview of how Imitation learning from Language Feedback (ILF) improves a base LLM $\pi_\theta$ via learning from natural language feedback. After $\pi_\theta$ generates multiple outputs $y_1, y_2, y_3$ for a given task, human annotators provide natural language feedback about how the outputs can be improved. Another model, $\pi_{\text{Refine}}$, intakes both the original outputs and the natural language feedback, and generates *refinements* of $y_1, y_2, y_3$ that incorporate the feedback. The higher quality refinements are then used to fine-tune $\pi_\theta$.

# 1 Introduction

Despite achieving strong performance across a variety of NLP tasks (*e.g.* summarization, question answering, and dialog (Radford & Narasimhan, 2018; Radford et al., 2019; Brown et al., 2020; Rae et al., 2021, *inter alia*)), large language models (LLMs) often generate text that violates human preferences. Examples of these violations include misinformation (Lin et al., 2021), offensive language (Gehman et al., 2020), factually incorrect summaries (Stiennon et al., 2020), and buggy or insecure code (Chen et al., 2021; Kang et al., 2022).

We hypothesize that these failures can be attributed in large part to modern LLM pre-training set-ups. For instance, modern pre-training datasets often consist mostly of minimally filtered text scraped from the Internet, which often contains insecure (Kang et al., 2022) or buggy (Chen et al., 2021) code and toxic text (Gao et al., 2020). This training signal also consists exclusively of offline demonstrations, without any signal from trial-and-error or interactive guidance that penalizes the model's buggy outputs. As such, we hypothesize that supervising LLMs with explicit human-written feedback on the model's own outputs can be more effective at training models to produce functionally correct code.

In particular, an intuitive and rich form of feedback to provide to LLMs is *natural language* feedback. We argue that LLMs are naturally able to incorporate written feedback, which has been shown to significantly improve a code generation model's pass rates when the feedback is provided at test time (Nijkamp et al., 2022; Austin et al., 2021). In our work, we build upon this observation by exploring the use of natural language feedback during the training process itself, rather than just during inference. We conjecture that such feedback provides expressive and targeted information about a code generation model's current failings in a sample-efficient manner. More broadly, this approach also represents a weak version of *scalable oversight* (Amodei et al., 2016; Bowman et al., 2022), in that model overseers can improve a model merely by evaluating its outputs, without manually generating new demonstrations, in a way that takes advantage of the capabilities that are being supervised. To train LLMs with language feedback, we propose an algorithm called Imitation learning from Language Feedback (ILF), which extends our previous non-archival work in Scheurer et al. (2022), who study the impact of learning from language feedback on text summarization models. Scheurer et al. (2022) improves a summarization model by training the base model on improved summaries generated from the model's original summaries and human-written feedback. Our work builds upon Scheurer et al. (2022) in a number of ways: (1) by formalizing the algorithm and generalizing it into

a form that can be applied to any task (our ILF algorithm in Section 2); (2) by detailing how the reward function can be adapted for both text summarization (Section 4.1) and code generation (Section 3.1); (3) by demonstrating a proof-of-concept of ILF for code generation; and (4) by scaling up our work on ILF for summarization via crowdsourced human annotations and conducting extensive experiments demonstrating the effectiveness of our method against other baselines.

ILF improves the quality of outputs generated by a baseline model $\pi_\theta$ by training a separate model $\pi_{\text{Refine}}$ to use language feedback to revise the low-quality outputs. (We refer to the revised outputs as *refinements*.) We then improve $\pi_\theta$ by fine-tuning it on the $\pi_{\text{Refine}}$-generated refinements that are of sufficiently high quality, yielding a final improved model $\pi_{\theta^*}$. The high-level algorithm is shown in Figure 1. This procedure may be run iteratively to continue improving the model, which we show can be seen as minimizing the expected KL divergence from a target ground truth distribution (Section 2).

We demonstrate the efficacy of ILF on two tasks – a proof-of-concept for code generation, and a larger scaled-up study of ILF for text summarization. For code generation, we show that ILF improves a CODEGEN-MONO 6.1B model's pass@1 rate on the Mostly Basic Python Problems (MBPP) benchmark (Odena et al., 2021) from 22% (after being fine-tuned on the ground truth programs provided by the MBPP dataset) to 36%. It also significantly outperforms zero-shot performance ($26\% \rightarrow 36\%$, see Section 3.3). For text summarization, we show that ILF improves LM-generated summaries monotonically with the amount of feedback provided (with experiments of up to 5K samples of feedback). In all data regimes, ILF leads to comparable or better results than fine-tuning on *human-written* summaries, suggesting that our approach is a strong alternative to supervised learning on human demonstrations. We also show that combining ILF with best-of-N sampling, where the samples are ranked by a model trained from binary comparison feedback, results in even higher quality summaries that are comparable to the quality of human-written summaries. Taken together, these results establish that learning from language feedback is a promising avenue for training LLMs.

## 2 High-Level Method

Here, we formally describe the problem we aim to tackle before introducing our algorithm.

### 2.1 Preliminaries

Suppose we start with vocabulary $\mathcal{V}$ and a pre-trained autoregressive language model $\pi_\theta$ parameterized by $\theta$. $\pi_\theta : \mathcal{V}^* \rightarrow [0,1]$ is a probability distribution over sequences of tokens $x \in \mathcal{V}^*$, where $\mathcal{V}^*$ is the set of all finite concatenations of tokens in $\mathcal{V}$. We also have a distribution of tasks $t \sim p_T$ and a reward function $R(x, t)$ which outputs some real-valued reward that encodes how high-quality a particular output $x$ is for task $t$. Lastly, we define a fine-tuning function FINETUNE($\pi_\theta, \mathcal{D}$) that applies a gradient-based optimization algorithm to $\pi_\theta$ using the associated loss objective calculated over dataset $\mathcal{D}$.

### 2.2 Imitation Learning from Language Feedback

Our high-level goal is to sample diverse outputs $x_1 \sim \pi_\theta(\cdot|t)$ for any given task $t$ sampled from the task distribution. We do so by fitting $\pi_\theta$ to approximate a ground truth distribution $\pi_t^*(x_1)$ that assigns a probability to $x_1$ that is proportional to its quality, as measured by the reward function $R$. Fitting $\pi_\theta$ to approximate $\pi_t^*$ can be seen as minimizing the expected KL divergence from $\pi_t^*$ to $\pi_\theta$ over the task distribution $p_T$:

$$\min_\theta \mathop{\mathbb{E}}_{t \sim p_T} \left[ \text{KL}(\pi_t^*, \pi_\theta(\cdot \,|\, t)) \right] \tag{1}$$

where

$$\pi_t^*(x_1) \propto \exp\left(\beta R(x_1, t)\right). \tag{2}$$

Minimizing the objective in Equation 1 is equivalent to supervised learning, *i.e.* minimizing the cross-entropy loss:

$$\mathcal{L}(\theta) = - \mathop{\mathbb{E}}_{t \sim p_T} \left[ \mathcal{L}_\theta(t) \right], \tag{3}$$

where

$$\mathcal{L}_\theta(t) = \sum_{x_1} \pi_t^*(x_1) \log \pi_\theta(x_1|t). \tag{4}$$

Rather than computing this loss over the exponentially large space of all possible $x_1$'s, we instead use Monte-Carlo sampling over a small set of $x_1$'s drawn from $\pi_t^*$. However, this is still intractable because we cannot sample directly from $\pi_t^*$. Instead, we approximate $\pi_t^*$ using importance sampling with a proposal distribution $q_t(x_1)$:

$$\mathcal{L}_\theta(t) = \sum_{x_1} q_t(x_1) \frac{\pi_t^*(x_1)}{q_t(x_1)} \log \pi_\theta(x_1|t) \tag{5}$$

which assigns higher weights to higher quality programs $x_1$.

## 2.3 Proposal Distribution $q$

Intuitively, we aim to design $q_t$ to be as close as possible to $\pi_t^*$, which we accomplish by incorporating pieces of natural language feedback $f$ that give information about how to transform a low-reward program $x_0$ into a higher-reward program $x_1$. This can be achieved by: (i) identifying a low-quality output $x_0 \sim \pi_\theta(\cdot|t)$ that currently has low reward $R(x_0, t)$, (ii) asking for natural language feedback $f$ about how to improve $x_0$, (iii) using $f$ to transform the original output $x_0$ into a *refinement* $x_1$ that incorporates the feedback and has reward $R(x_1, t) > R(x_0, t)$, and (iv) assigning higher weight to $x_1$.

We can formalize this procedure as follows. Let $\pi_\psi(x_1|t, x_0, f)$ be a distribution over outputs $x_1$ that improve $x_0$ by incorporating the feedback $f$ and $p_\mathcal{F}(f \,|\, t, x_0)$ be the distribution of pieces of feedback $f$ for output $x_0$ and task $t$. We can then define our proposal distribution as

$$q_t(x_1) = \sum_{x_0, f} \pi_\theta(x_0|t) \times p_\mathcal{F}(f|t, x_0) \times \pi_\psi(x_1|t, x_0, f), \tag{6}$$

which results from marginalizing $q_t(x_1)$ over all pairs of $(x_0, f)$ and factorizing the joint distribution of $x_1, x_0$, and $f$ given $t$.

In some tasks (*i.e.* summarization), we assume that every initial output $x_0 \sim \pi_\theta(\cdot|t)$ is low-quality and can be improved upon (Section 4). However, for other tasks (*i.e.* code generation), we may be able to identify initial outputs that are already high-quality and do not need feedback. For example, if the code is already functionally correct. In this case, we specify an additional *quality function* $\phi(x, t)$ such that

$$\phi(x, t) := \begin{cases} 1, & \text{if } R(x, t) \geq \gamma \\ 0, & \text{otherwise} \end{cases} \tag{7}$$

for some reward threshold $\gamma$. We can then modify Equation (6) to place all support on pairs of $(x_0, x_1)$ where $\phi(x_0, t) = 0$ and $\phi(x_1, t) = 1$:

$$q_t^{\text{Filtered}}(x_1) = \sum_{x_0, f} \pi_\theta(x_0|t) \times \delta_0\left(\phi(x_0, t) \,|\, x_0, t)\right) \times p_\mathcal{F}(f|t, x_0) \times \pi_\psi(x_1|t, x_0, f) \times \delta_1(\phi(x_1, t) \,|\, t, x_1) \tag{8}$$

where we write $\delta_i(j)$ for the Kronecker delta distribution, where

$$\delta_i(j) = \begin{cases} 1, & \text{if } i = j \\ 0, & \text{if } i \neq j \end{cases}$$

To sample from $q_t^{\text{Filtered}}$ and $q_t$, we defer discussion to Sections 3 and 4 respectively.

Now that we have devised a way to sample from $q_t$, we can rewrite Equation 5 as

$$\mathcal{L}_\theta(t) = \mathop{\mathbb{E}}_{x_1^i \sim q_t} \underbrace{\frac{\pi_t^*(x_1^i)}{q_t(x_1^i)}}_{w_i} \log \pi_\theta(x_1^i|t), \tag{9}$$

where $w_i$ is the importance weight of the $i$-th sample from $q_t$. However, since we can only sample from $q_t$, we cannot directly compute $w_i$. Instead, we assume that $q_t(x_1^i)$ is relatively constant for all samples $x_1^i$ due to the high quality of human feedback. This allows us to simply compute the un-normalized value of $\pi_t^*(x_1^i)$ (from Equation 2) and use self-normalization instead. If we sample using low temperatures (resulting in $\beta \to \infty$), then this is approximately equivalent to computing the likelihood of only the best refinement $x_1^*$ for each task $t$, *i.e.*

$$\mathcal{L}(\theta) \approx - \mathop{\mathbb{E}}_{t \sim p_T} \left[ \log \pi_\theta(x_1^* | t) \right] \tag{10}$$

Notably, this objective is equivalent to supervised learning over the best refinements per task. Using natural language feedback simply allows us easier access to these refinements.

Our objective of approximating the ground truth distribution $\pi_t^*$, which is proportional to the reward $R$, has clear connections to maximizing reward in reinforcement learning (RL). However, the goal in RL is to find the best policy that maximizes the reward, whereas our algorithm results in a training dataset of high-quality outputs $x_1$ for task $t$, which allows us to train our model to generate a diverse set of outputs that achieve high rewards on average. The broad diversity of high-quality outputs endows downstream users and systems with more control over which aspects they prefer and want to avoid.

## 3 Code Generation

Program synthesis, the automatic generation of computer programs from an input specification (*e.g.* a natural language task description or a set of input-output examples) (Manna & Waldinger, 1971), is an important task for the field of software engineering. Effective program synthesis can not only improve the efficiency of software developers (Ziegler et al., 2022), but also increase the accessibility of writing code in general. Recently, pre-trained large language models (LLMs) have demonstrated impressive success on program synthesis (Chen et al., 2021; Li et al., 2022a; Austin et al., 2021; Nijkamp et al., 2022; Xu et al., 2022a, *inter alia*) but still struggle to consistently generate correct code, even with large-scale pre-training (Chen et al., 2021). We select code generation as a testbed for our approach not only due to its importance but also due to the ready availability of a precise reward function (*i.e.* functional correctness of the code) and evidence that current state-of-the-art code generation models still perform poorly on difficult programming challenges (OpenAI, 2023).

### 3.1 ILF for Code Generation

In code generation, each task consists of pairs $(t, u) \in \mathcal{T}$ of a natural language task description $t \in \mathcal{T}$ (*e.g.* "Write a function that computes the prime factorization of an input integer.") and a suite $u = \text{UNITTESTS}(t) \in \mathcal{U}$ of unit tests associated with task $t$. Our reward function $R$ is a unit test verification function $\text{EVAL} : \mathcal{V}^* \times \mathcal{T} \to \{0, 1\}$ that indicates whether a program $x \sim \pi_\theta(\cdot | t)$ passes all the unit tests in $\text{UNITTESTS}(t)$:

$$\text{EVAL}(x, t) \coloneqq \begin{cases} 1, & \text{if } x \text{ passes test suite} \\ & \quad \text{UNITTESTS}(t), \\ 0, & \text{otherwise} \end{cases} \tag{11}$$

We use $\text{EVAL}(x, t)$ as our reward function $R(x, t)$ and the identity function as our quality function $\phi(x, t)$. It follows that Equation 8 can be adapted for code generation as follows:

$$q_t^{\text{Filtered}}(x_1) = \sum_{x_0, f} \left[ \pi_\theta(x_0 | t) \times \delta_0 \left( \text{EVAL}(x_0, t) \,|\, x_0, t) \right) \times p_{\mathcal{F}}(f | t, x_0) \times \pi_\psi(x_1 | t, x_0, f) \right.$$
$$\left. \times \delta_1 (\text{EVAL}(x_1, t) \,|\, t, x_1) \right] \tag{12}$$

This proposal distribution is guaranteed to place higher probability mass on higher-quality programs (in terms of unit test pass rate) than $\pi_\theta$ since the term $\delta_1(\text{EVAL}(x_1, t) \,|\, t, x_1)$ equals 0 for incorrect programs $x_1$.

We approximate sampling from $q_t^{\text{Filtered}}$ by considering each of the terms in Equation 12 in order:

1. We first sample from $\pi_\theta(x_0|t) \times \delta_0\left(\text{EVAL}(x_0,t) \,|\, x_0, t\right)$ by rejection sampling from $\pi_\theta$. In other words, we sample programs $x_0$ from $\pi_\theta$ for task $t$ and only keep those that fail the test suite (*i.e.* $\text{EVAL}(x_0,t) = 0$; step 2 of Algorithm 1).

2. We approximate sampling from $p_{\mathcal{F}}(f|t, x_0, \text{EVAL}(x_0,t) = 0)$ by having humans annotate programs $x_0$ (paired with their corresponding task descriptions $t$ and test suites $u$) with natural language feedback (step 3 of Algorithm 1).

3. We approximate sampling from $\pi_\psi(x_1|t, x_0, f)$ by sampling from $\pi_{\text{Refine}}$, a model capable of generating refinements given the task description, original programs, and human-written feedback.

4. Finally, the term $\delta_1(\text{EVAL}(x_1,t) \,|\, t, x_1)$ corresponds to another filter: we only keep refined programs $x_1$ that pass the test suite.

The complete algorithm is summarized in Figure 2 and Algorithm 1. Next, we consider more concrete details of how this sampling is accomplished.

**Using $\pi_{\text{Refine}}$ to Incorporate Feedback** ILF assumes the availability of feedback but not necessarily of the repaired code/refinements, for a variety

Figure 2: An overview of imitation learning from language feedback (ILF) for code generation. Given an initial LLM $\pi_\theta$, we sample programs from $\pi_\theta$ that do not pass unit tests (indicated by the red X). Human annotators write natural language feedback for the incorrect program and a model $\pi_{\text{Refine}}$ generates a *refinement* - *i.e.* an improved version of the original program that incorporates the feedback and passes the unit tests. Finally, we fine-tune $\pi_\theta$ on the refinements.

of reasons. We assume that program synthesis may be a task for which writing high-level natural language feedback is often less laborious than performing program repair. Although writing feedback involves identifying at a high level what is wrong with the program and how it should be fixed, program repair may involve the additional steps of refactoring, looking through documentation, and testing. Moreover, past work (Austin et al., 2021; Nijkamp et al., 2022) has indicated that certain large LLMs can proficiently incorporate the feedback at inference time, assuming access to accurate and high-quality feedback. As such, ILF assumes access to some model $\pi_{\text{Refine}}$ that is capable of producing a refinement given the original program and feedback. In many cases, $\pi_{\text{Refine}}$ can perform this task via few-shot prompting, without additional training (Nijkamp et al., 2022). We explore such possibilities in Appendix A.4. As such, ILF for code generation does not necessarily require extra training data for $\pi_{\text{Refine}}$.

---

**Algorithm 1** A single round of imitation learning from natural language feedback for code generation.

1: **Input:** Dataset $\mathcal{D}$, initial LLM $\pi_\theta$, unit test verification function EVAL, LLM $\pi_{\text{Refine}} : \mathcal{V}^* \to [0,1]$ trained to incorporate feedback into code
2: $C \leftarrow \{(x_0, t, u) \,|\, x_0 \sim \pi_\theta(\cdot|t), \text{EVAL}(x_0, t) = 0, (t, u) \in \mathcal{D}\}$
3: $C_{\text{annotated}} \leftarrow \{(x_0, f, t) \,|\, (x_0, t, u) \in C\}$       ▷ Humans write feedback $f$ for $x_0 \in C$.
4: $\mathcal{D}_{\text{Refined}} \leftarrow \{(t, x_1) \sim \pi_{\text{Refine}}(\cdot \,|\, t, x_0, f) \,|\, \text{EVAL}(x_1, t) = 1, (x_0, f, t) \in C_{\text{annotated}}\}$   ▷ $\pi_{\text{Refine}}$ generates refinements $x_1$ that incorporate feedback $f$ into $x_0$.
5: $\pi_{\theta*} \leftarrow \text{FINETUNE}(\pi_\theta, \mathcal{D}_{\text{Refined}})$

---

$\pi_{\text{Refine}}$ can take a variety of forms, but due to constraints on available funding and compute resources, we fine-tune a pre-trained CODEGEN-MONO 6.1B model as our $\pi_{\text{Refine}}$.[1] For the fine-tuned $\pi_{\text{Refine}}$, we create a training dataset by further annotating a subset of $C_{\text{annotated}}$ with refinements $x_1$ that repair incorrect programs $x_0$ by incorporating feedback $f$, such that $\text{EVAL}(x_1, t) = 1$ for $(x_0, f, t) \in C_{\text{annotated}}$. Further details of our dataset and annotation procedure are in Section 3.2.

## 3.2 Experiments and Results

Having described our high-level approach, we now explain the experimental setup we use to test ILF for code generation. Our data and code are open-sourced at `https://github.com/nyu-mll/ILF-for-code-generation`.

**Dataset** We train and evaluate our models on the Mostly Basic Python Problems (MBPP) dataset (Odena et al., 2021). MBPP contains 974 Python programming tasks designed to be solvable by entry-level coders. Each task contains a natural language task description $t$ (*e.g.*, "Write a function to return the prime factorization of the input."), a ground truth solution, and a suite $u$ of three unit tests. Since the task descriptions are sometimes ambiguous, we include one unit test in the task description. The addition of the unit test helps to specify the input and output format of each task. We hold out the remaining unit tests for the evaluation of our generated programs.

MBPP includes a designated prompt/training/validation/test split of the dataset, but we re-split the dataset into the following splits:

- MBPP$_{\text{Refine}}$: These are tasks with IDs in the range 111-310 for which CODEGEN-MONO 6.1B did not generate any correct completions. For the experiments where $\pi_{\text{Refine}}$ is a fine-tuned model, this split is used to train $\pi_{\text{Refine}}$.

- MBPP$_{\text{Train}}$: These are tasks with IDs in the range 311-974 for which CODEGEN-MONO 6.1B did not generate any correct completions. This split is first used to evaluate the correctness of refinements generated by $\pi_{\text{Refine}}$. Then, the correct refinements in this split are used to train $\pi_\theta$ to obtain $\pi_{\theta^*}$ (step 5 in Algorithm 1).

- MBPP$_{\text{Test}}$: These are tasks with IDs in the range 11-110 that we use to evaluate the final performance of $\pi_{\theta^*}$. Unlike the previous two splits, we use *all* tasks in this split, rather than only the tasks for which CODEGEN-MONO 6.1B did not originally generate correct programs for. This allows us to better compare the baseline performance of $\pi_\theta$ with that of $\pi_{\theta^*}$.

We use this modified split so that a larger portion of the dataset can be used to train the final model $\pi_{\theta^*}$, whereas smaller portions are allocated for training $\pi_{\text{Refine}}$ and evaluating $\pi_{\theta^*}$. We do not make use of the prompt split (IDs 1-10).

**Models** Throughout this paper, we use a pre-trained CODEGEN-MONO 6.1B model (Nijkamp et al., 2022) as our $\pi_\theta$. It is pre-trained sequentially on THEPILE (Gao et al., 2020), BIGQUERY (Nijkamp et al., 2022), and BIGPYTHON (Nijkamp et al., 2022). We selected this model because it is open-source, can be fine-tuned on a single $4 \times 100$ A100 (80 GB) node, and demonstrated pass@k scores comparable to CODEX-12B (Chen et al., 2021; Nijkamp et al., 2022).

To implement our algorithm, we independently fine-tune two separate instances of CODEGEN-MONO 6.1B to create $\pi_{\text{Refine}}$ and the final model $\pi_{\theta^*}$. We train $\pi_{\text{Refine}}$ using pairs of incorrect programs and human-written feedback as inputs, with human-written refinements as targets (using the format in Figure 7). In contrast, we train $\pi_{\theta^*}$ using natural language task descriptions from MBPP as the inputs and $\pi_{\text{Refine}}$-generated refinements as the targets. Further training details are in Appendix A.2.

---

[1] We also investigate using few-shot `gpt-3.5-turbo` and `gpt-4` as our $\pi_{\text{Refine}}$ in Appendix A.4, to demonstrate that ILF does not necessarily require training a custom $\pi_{\text{Refine}}$ model. Oftentimes, prompting an off-the-shelf LLM suffices.

**Evaluation**  We evaluate all code generations in this paper using the *pass@k* metric introduced in Kulal et al. (2019). It estimates the rate for which $\geq 1$ of $k$ model samples passes all the unit tests. We use the empirical estimate of this quantity from Chen et al. (2021), an unbiased estimator given by:

$$\text{pass@k} := \mathbb{E}_{\text{task}} \left[ 1 - \frac{\binom{n-c}{k}}{\binom{n}{k}} \right] \tag{13}$$

for $n$ total programs (where $n \geq k$) and $c$ correct programs for the given task.

**Human Annotation**  We hire annotators via Surge AI[2] to write both natural language feedback and refinements for incorrect programs generated by CodeGen-Mono 6.1B. For each task that CodeGen-Mono 6.1B generated no correct programs for, we ask the workers to first select one of the incorrect programs to write feedback and refinement for. We specify that the workers should select a sample that seems relatively easy to correct (*i.e.* could be minimally corrected to pass the unit tests). Then, they are asked to write feedback that describes what is wrong with the current code and how to fix it. For the refinement, they are asked to copy over the original code and make the *minimum number of edits necessary* to incorporate the feedback and pass all the unit tests. The full set of worker instructions can be found in Appendix A.3.

We keep all annotations for which the refinement passes all tests in the task's test suite, the feedback is correct (as manually verified by the authors), and the Levenshtein edit distance between the refinement and the original program is less than 50% of $\max(\text{len}(\text{refinement}), \text{len}(\text{original program}))$. The final dataset consists of 195 triples of (incorrect program, human-written feedback, human-written refinement). On average, workers are paid $23 per annotated sample and take 27 minutes/sample, with a 10th percentile of 4 minutes and a 90th percentile of 43 minutes.

Although the ILF algorithm only requires the collection of human-written feedback for the tasks in $\text{MBPP}_{\text{Train}}$ (assuming access to some $\pi_{\text{Refine}}$ that is already fine-tuned or can generate refinements via few-shot prompting), we collect both human-written feedback and refinement for all splits of the data so that we can conduct further analyses of our method. For instance, this allows us to compare fine-tuning on $\pi_{\text{Refine}}$-generated refinements with fine-tuning on human-written refinements. When scaled to other pairs of model and task, ILF requires new feedback annotations, but it is possible that using ILF on one dataset will improve the model's abilities on another dataset for a similar task. We leave analyses of further scaling ILF across different tasks and models to future work.

Table 1: Initial zero-shot CodeGen-Mono 6.1B performance on the entire MBPP dataset. "1+ Correct" refers to the percentage of tasks for which CodeGen-Mono 6.1B generated at least one program that passed all unit tests.

| Metric | Zero-Shot CodeGen-Mono 6.1B |
|---|---|
| Pass@1 | 31% |
| Pass@10 | 63% |
| 1+ Correct | 67% |

**CodeGen-Mono 6.1B Incorporates Feedback**  We first verify that our baseline model can use feedback to repair incorrect code, a prerequisite for ILF to work. We evaluate CodeGen-Mono 6.1B's ability to generate refinements given pairs of (incorrect code, natural language feedback), both in a few-shot manner and after fine-tuning. Feedback is only required for tasks for which $\pi_\theta$ is initially unable to produce a correct response, so we first evaluate CodeGen-Mono 6.1B zero-shot on all of MBPP, generating 30 programs per task with temperature 0.8. Table 1 shows the resulting pass rates. There were 321 tasks for which zero-shot

---

[2]www.surgehq.ai

Table 2: Evaluations of 1-shot refinements generated by CODEGEN-MONO 6.1B (before ILF) given either related or unrelated text feedback in the prompt. Feedback is provided only for tasks on which CODEGEN-MONO 6.1B previously did not output any correct programs.

| Prompt Type | CODEGEN-MONO 6.1B | |
| --- | --- | --- |
| | Pass@1 ↑ | Pass@10 ↑ |
| Code + feedback | 2.0% | 13.8% |
| Code + unrelated feedback | 0.4% | 4.0% |

CODEGEN-MONO 6.1B yielded no correct samples (from Table 1: $(100\% - 67\%) \times 974$ tasks $\approx 321$). We then annotate one incorrect program per task with both feedback and refinement, as described in Section 3.2.

**Few-Shot Feedback Incorporation** We use the human feedback annotations to create few-shot feedback prompts, formatted as in Figure 7. We evaluate CODEGEN-MONO 6.1B's ability to produce refinements that incorporate the feedback and pass the unit tests. However, producing a refinement that passes the unit tests does not guarantee that the feedback has been incorporated; there can be multiple solutions to a programming task, including ones that are functional but completely different and not using the feedback to improve upon the original code. Alternatively, the model may already be able to repair programs without feedback. Thus, we also evaluate the pass rate after shuffling the feedback samples in the dataset, to evaluate if the model's ability to repair code degrades when presented with unrelated feedback.

The results are shown in Table 2. CODEGEN-MONO 6.1B's ability to incorporate relevant feedback on this particular set of program is low, with pass@10 reaching only 13.8%. However, the gap in accuracy between CODEGEN-MONO 6.1B-generated refinements on relevant versus irrelevant feedback is significant, with pass@10 decreasing by 71% (relative; $13.8\% \rightarrow 4.0\%$), indicating that the model is indeed using the feedback.

**Training $\pi_{\text{Refine}}$** Next, we examine whether we can improve our ability to repair programs given feedback by fine-tuning a separate model specifically to perform this task. As mentioned prior in Section 3.1, ILF does not strictly require training a separate model for $\pi_{\text{Refine}}$. A large pre-trained language model may be able to refine the model outputs in a few-shot manner, without further training. However, due to the closed nature of some models and the amount of compute and API funds required for larger models, we also choose to fine-tune our own $\pi_{\text{Refine}}$. Our training examples consist of triples of incorrect program, human-written feedback, and human-written refinement. We train the model to maximize the likelihood of the refinement given the program and feedback. The incorrect programs are generated by CODEGEN-MONO 6.1B zero-shot on MBPP tasks, and the feedback and refinements are written by human annotators, as discussed in Section 3.2. We only include tasks for which none of CODEGEN-MONO 6.1B's generated programs are correct, yielding 44 tasks in the training dataset (forming the split MBPP$_{\text{Refine}}$) and 128 tasks in the evaluation dataset (forming the split MBPP$_{\text{Train}}$). We ask human annotators to write refinements of the original code that incorporated their own previously written feedback, passed the unit tests, and make only minimal edits to the code (see Section 3.2). The format of the training data also matches the few-shot prompt format (Figure 7) but without the in-context examples of refinements. We denote this model as $\pi_{\text{Refine}}$, as described in Section 2.3.

Table 3 shows the pass rates for $\pi_{\text{Refine}}$ on MBPP$_{\text{Train}}$, which were produced by sampling 30 refinements per task with temperature 0.8. Fine-tuning significantly improves CODEGEN-MONO 6.1B's ability to incorporate feedback compared to 1-shot refinement, increasing pass rates more than three-fold ($2\rightarrow19\%$ pass@1, $13.8\rightarrow47\%$ pass@10, from Tables 2 and 3). Furthermore, 61% of tasks had at least one correct refinement. This is particularly significant when considering the fact that we selected only tasks for which a non-fine-tuned CODEGEN-MONO 6.1B model did not originally output any correct programs for (the rightmost column in Table 3). For the 61% of validation tasks that $\pi_{\text{Refine}}$ generated a correct refinement for, we randomly selected one such correct program for each task to form the training dataset for our final model $\pi_{\theta*}$, yielding

Table 3: Pass rates of $\pi_{\text{Refine}}$-generated refinements versus zero-shot CODEGEN-MONO 6.1B programs for tasks in MBPP$_{\text{Train}}$. MBPP$_{\text{Train}}$ contains only tasks for which CODEGEN-MONO 6.1B did not generate any correct completions.

| Metric | $\pi_{\text{Refine}}$ | Zero-shot CODEGEN-MONO 6.1B |
|---|---|---|
| Pass@1 | 19% | 0% |
| Pass@10 | 47% | 0% |
| 1+ correct | 61% | 0% |

a final training dataset of 78 examples. However, we show in Appendix A.6 that $\pi_{\text{Refine}}$'s ability to generate correct refinements declines monotonically as the number of bugs addressed in the feedback increases.

Table 4: Final performance of $\pi_{\theta*}$ on MBPP$_{\text{Test}}$, compared to other ablations and baselines. All results are calculated using 30 output samples with temperature 0.8. All the methods are built on the CODEGEN-MONO 6.1B model.

| Method | Feedback Source | Fine-Tuning Data | Pass Rates of $\pi_{\theta*}$ | |
|---|---|---|---|---|
| | | | Pass@1 | Pass@10 |
| ILF | Humans | $\pi_{\text{Refine}}$ refinements | **36%** | **68%** |
| Ablations | 1-shot InstructGPT | 1-shot InstructGPT refinements | 19% | 55% |
| | 2-shot InstructGPT | 2-shot InstructGPT refinements | 25% | 59% |
| Gold Standard | - | Human-written refinements | 33% | **68%** |
| Baseline | - | MBPP ground truth programs | 22% | 63% |
| Zero-Shot | - | - | 26% | 59% |

## 3.3 ILF Yields Pass Rates Higher Than Fine-Tuning on Ground Truth Data or Human-Written Programs Alone

Given that our refinements improve over the initial programs, we now fine-tune on the refinements to improve our code generation model. As discussed earlier, we use the correct refinements (as evaluated by the unit tests) that $\pi_{\text{Refine}}$ generated for its evaluation dataset as the training dataset for $\pi_{\theta*}$. Since $\pi_{\theta*}$ is meant to generate code from a natural language task description (rather than to incorporate feedback into a refinement), the inputs of our training dataset are the MBPP prompts and the targets are the 78 $\pi_{\text{Refine}}$-generated refinements described in the previous section. We also compare the performance of $\pi_{\theta}^{*}$ against that of CODEGEN-MONO 6.1B evaluated in a zero-shot manner, CODEGEN-MONO 6.1B fine-tuned on the ground truth programs from the MBPP dataset, and CODEGEN-MONO 6.1B fine-tuned on our human-written refinements. In realistic use cases, human-written refinements are not readily available, so we consider fine-tuning on this dataset to be a **gold standard**. ILF does not require ground truth programs or human-written refinements – we merely use them here for evaluation. For all fine-tuning experiments, we train on programs corresponding to the same set of task IDs as the ones used in $\pi_{\theta*}$'s training dataset.

Additionally, we evaluate the impact of ablating the human annotations in our algorithm by using an LLM in place of humans to generate the feedback and refinements (replacing steps 3 and 4 in Algorithm 1). For the LLM, we use GPT-3.5 fine-tuned with Feedback Made Easy (FeedME; `text-davinci-002` on the OpenAI API)[3]. We refer to this model as InstructGPT, which is the series of OpenAI models that FeedME belongs to (OpenAI, 2022b). We use InstructGPT to generate both the feedback and refinements on the original programs. We use the same prompt format as for $\pi_{\text{Refine}}$ (Figure 7), with two slight modifications: (1) the feedback prompt ends at "Feedback:" whereas the refinement prompt uses the entire prompt (with the previously InstructGPT-generated feedback inserted after "Feedback:"), and (2) we use $k$ in-context

---

[3]Details at beta.openai.com/docs/model-index-for-researchers

examples, with $k$ indicated in Table 4. We then fine-tune CODEGEN-MONO 6.1B on the model-generated refinements.

The results of our ILF algorithm compared to the baselines and ablations are shown in Table 4. ILF yields the highest pass@1 and pass@10 rates, despite how few samples of feedback and refinements we use. The pass@1 rate in particular shows a significant increase in improvement over the baseline of fine-tuning on MBPP ground truth programs, increasing from 22% to 36%. Pass@1 improvements are especially helpful for assisting with software engineering, where it is more helpful to suggest a single correct completion rather than 10 possible completions for the user to select from.

ILF also outperforms both zero-shot inference and fine-tuning on human-written refinements on the pass@1 metric, yielding increases of $26\% \rightarrow 36\%$ and $33\% \rightarrow 36\%$ in pass@1 rates, respectively. However, training on human-written refinements yielded comparable pass@10 rates as ILF, which is unsurprising since $\pi_{\text{Refine}}$ was trained on human-written refinements. When human-written feedback and $\pi_{\text{Refine}}$-generated refinements are ablated (the "Ablations" section of Table 4), ILF also outperforms training on both 1-shot and 2-shot InstructGPT-generated refinements, with increases of $19\% \rightarrow 36\%$ and $25\% \rightarrow 36\%$, respectively.

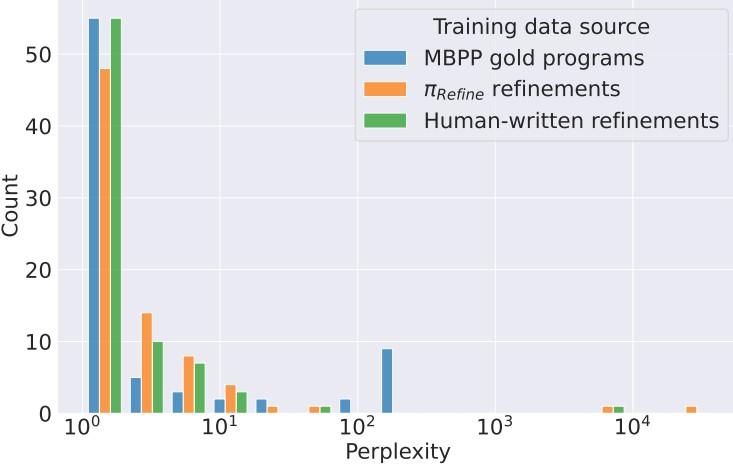

Figure 3: Histogram of the perplexities of the various training data sources, as measured using a pre-trained CODEGEN-MONO 6.1B model.

**Analysis of Training Data Sources**  However, we also note the surprising fact that merely training on a small sample of the MBPP gold programs did not make a significant difference in accuracy over zero-shot inference. We speculate that the gold programs from the MBPP dataset may be somewhat out-of-distribution for CODEGEN-MONO 6.1B. To test this hypothesis, we computed the perplexity of the MBPP gold programs, the $\pi_{\text{Refine}}$-generated refinements, and the human-written refinements using the pre-trained CODEGEN-MONO 6.1B model. The results are shown in Figure 3. While the distributions of all three data sources look similar, the MBPP dataset contains more high-perplexity programs (*i.e.* programs with perplexity $\geq 10^2$) than either the $\pi_{\text{Refine}}$-generated refinements or the human-written refinements. As a result, it is likely easier for CODEGEN-MONO 6.1B to learn from the latter two datasets, since they are closer to CODEGEN-MONO 6.1B's original distribution while still being functionally correct.

Furthermore, ILF is particularly useful for settings where large amounts of gold code are not available. In this setting, ILF can be thought of as a method of not only generating more training data, but training data that is closer to the model's original outputs in data representation space and that specifically repairs the kinds of bugs that the original model generates. As a result, fine-tuning the model on $\pi_{\text{Refine}}$-generated refinements does not require adjusting the weights as much as fine-tuning the model on the MBPP gold programs would, even though both training datasets contain the same number of functionally correct programs.

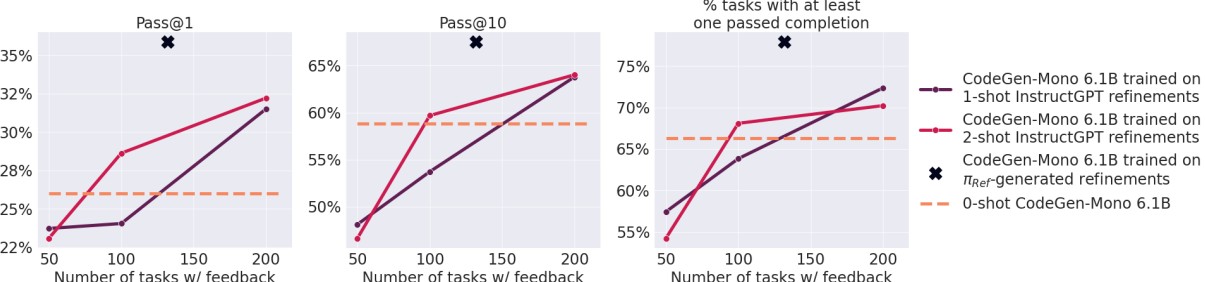

Figure 4: Training dataset size versus CODEGEN-MONO 6.1B pass rates on MBPP tasks 11-111 after fine-tuning on InstructGPT-generated refinements, versus the performance of $\pi_{\theta*}$ (the model produced by our approach). $X$ marks the performance of $\pi_{\theta*}$, whereas the solid lines plot the performance of CODEGEN-MONO 6.1B after fine-tuning on correct refinements generated by InstructGPT, using feedback also generated by InstructGPT. The dashed line indicates the zero-shot pass rate of a pre-trained CODEGEN-MONO 6.1B model.

### 3.4  Scaling Up Model Feedback Does Not Offer the Same Benefits As Human Feedback

Since high quality human feedback can be expensive to collect, we also evaluated how much model feedback might yield the same benefit as our sample of human-written feedback. To do so, we randomly select $k$ tasks from the set of MBPP tasks for which CODEGEN-MONO 6.1B did not originally output a correct answer, and prompt InstructGPT to generate both the feedback and the refinement. We then evaluate the refinements for correctness and train CODEGEN-MONO 6.1B on the correct refinements. We use $k \in \{50, 100, 200\}$ and generate 30 output samples at temperature 0.8 for all stages of the experiment. We are limited to these $k$ values due to the small number of tasks we have in MBPP$_{\text{Train}}$, but future work may investigate scaling up these experiments by using larger datasets or automatically generating new tasks and unit tests for the training dataset. Further training details are listed in Appendix A.2.

The results are shown in Figure 4. Although increasing the quantity of InstructGPT-generated feedback offers modest improvements in pass rates, these improvements do not yield pass rates as high as those of $\pi_{\theta*}$, even though $\pi_{\theta*}$ uses only a total of 122 pieces of feedback throughout its training process (44 for training $\pi_{\text{Refine}}$ and 78 for generating refinements to train $\pi_{\theta*}$ on). However, as pre-trained large language models continue to improve dramatically in quality, we expect that this gap between human- and model-written feedback will increasingly narrow.

We also analyze some of the qualitative differences between human-written and model-generated feedback in Appendix A.5 and provide some specific examples in Appendix A.7. In summary, model-generated feedback is more likely to be less correct, less relevant, and less informative (*i.e.* addresses fewer bugs) than human-written feedback.

## 4  Summarization

We also adapt and evaluate ILF on the real-world task of text summarization. Applying ILF to summarization differs from applying it to code generation in a number of ways. Unlike code generation, text summarization has no automated method for evaluation that is completely precise. Although automated metrics such as ROUGE, BLEU, and METEOR exist, past work has shown that these metrics often do not correlate highly with human-rated evaluations of summarization quality, especially on aspects related to coherence and relevance (Fabbri et al., 2021). As such, we utilize human evaluations to compare the performance of ILF versus baseline methods on text summarization. We also utilize an instruction-fine-tuned `text-davinci-001` model to rank the generated refinements and select which ones to train on.

Furthermore, we assume that all summaries generated by $\pi_\theta$ can be improved upon. This eliminates the need to filter for only low-quality initial outputs and simplifies the proposal distribution to Equation 6. Taken

together, the high-level algorithm for ILF applied to summarization is shown in Figure 5 and detailed in Algorithm 2.

## 4.1 ILF for Text Summarization

In text summarization, each task consists of a context $c$ (a source document) that we aim to generate improved outputs $x_1$ (e.g., high-quality summaries) for, according to human preferences. As before, $x_1$ are *refinements* of initial outputs $x_0$ that are generated via incorporating language feedback $f$. To implement the reward function $R$ (from Equation 2), we condition an instruction-fine-tuned LLM on a binary question such as *Does this new text [$x_1$] incorporate the feedback [$f$] provided on the initial text [$x_0$]? Answer Yes or No.*, where the label $y$ is either $y_{\text{good}}$ ("Yes") or $y_{\text{bad}}$ ("No").[4] Then if we let $\mathcal{I}$ be an indicator variable that indicates whether $x_1$ is a high-quality summary of $c$, we can approximate the reward function $R$ using the probability of the positive answer $y_{\text{good}}$, i.e.:

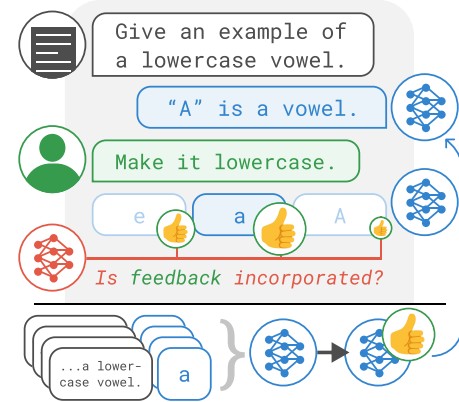

Figure 5: In ILF for text summarization, $\pi_{\text{Refine}}$ generates refinements of the original output based on the feedback. We use an LM to pick the best refinement and fine-tune the original LM to maximize the likelihood of the chosen refinement.

$$R(x_1, c) := \Pr(x_1 \text{ is a high-quality summary of } c) \tag{14}$$
$$= p(\mathcal{I}|x_1, c) \qquad\qquad \text{(By definition)} \tag{15}$$
$$= p(\mathcal{I}|x_1, c, x_0, f) \qquad\qquad \text{(Conditional independence)} \tag{16}$$
$$\approx \frac{p(y_{\text{good}}, x_0, f, x_1, c)}{p(y_{\text{good}}, x_0, f, x_1, c) + p(y_{\text{bad}}, x_0, f, x_1, c)} \tag{17}$$

where we assume that $\mathcal{I}, x_1$, and $c$ are conditionally independent given $x_0$ and $f$.

## 4.2 Dataset

We evaluate the effectiveness of ILF on the task of text summarization using the TL;DR dataset (Völske et al., 2017), which consists of Reddit titles, posts, and their corresponding summaries. Stiennon et al. (2020) adapt this dataset and show that it is a more realistic task for evaluating summarization models compared to the commonly used CNN/DM dataset (Hermann et al., 2015). To ensure the quality of our dataset, we follow the same preprocessing steps as outlined in Stiennon et al. (2020) and extract a train dataset with 5000 samples, a development dataset with 200 samples, a validation dataset with 500 samples, and a test dataset with 698 samples[5].

We then hire experienced annotators through Surge AI[6] to create our language feedback dataset, which we open source along with our code.[7] For each sample, we first generate three summaries for each Reddit post using the instruction-fine-tuned model `text-davinci-001`(FeedME) (Ouyang et al., 2022; OpenAI, 2022b). Two of these summaries are used for a binary comparison, in which annotators indicate their preference. The third summary serves as the initial output for which we solicit language feedback. This feedback should address the single most important shortcoming of the summary and can be related to coverage (how well the summary covers the important information in the post), accuracy (the factual accuracy of the summary), coherence (the coherence of the summary on its own), or other. We do not impose any restrictions on how the feedback should be written. In addition to providing feedback, annotators are also asked to write an ideal

---

[4]For the complete prompts, please see Appendix B.2.

[5]The train and development datasets are taken from Stiennon et al. (2020)'s train dataset, and the validation and test set are taken from their test dataset.

[6]https://surgehq.ai

[7]Data: https://huggingface.co/datasets/JeremyAlain/SLF5K; Code: https://github.com/JeremyAlain/imitation_learning_from_language_feedback

summary that is maximally 48 tokens long. The same crowd worker annotates all three tasks for a given sample. Overall, the dataset collection and human evaluations cost 40K$. On selected samples of the binary comparison task, we achieve an author-annotator agreement of 81.0% and annotator-annotator agreement of 70.0%. The human summaries we collect are of excellent quality, as demonstrated in a human evaluation, where we compare our human-written summaries to the ones automatically extracted from Reddit (Völske et al., 2017) (also used as baselines in Stiennon et al. (2020); Scheurer et al. (2022)). We find that our human-written summaries are preferred $72.0 \pm 3.2\%$ of the time, making them a much stronger baseline than the Reddit-sourced summaries.

---

**Algorithm 2** Imitation learning from natural language feedback for text summarization.

---

1: **Input:** Number of iterations $K$, sequence of sets of source documents $\mathcal{C} = [\mathcal{C}_1, ..., \mathcal{C}_K]$, language model $\pi_\theta$, refinement language model $\pi_{\text{Refine}}$, reward model $R$
2: **for** $k$ **in** $1...K$ **do**
3:     Initialize fine-tuning dataset $\mathcal{D}_k = \{\}$
4:     **for** document $c$ **in** $\mathcal{C}_k$ **do**
5:         $x_0 \sim \pi_\theta(x_0|c)$
6:         Human provides feedback $f$ on $(c, x_0)$
7:         $\{x_1^1, \ldots, x_1^N\} \sim \pi_{\text{Refine}}(x_1|c, x_0, f)$
8:         $x_1 = \text{argmax}_{x_1^i} R(x_1^i|x_0, f, c)$             ▷ The highest-reward refinement is selected.
9:         Add $(c, x_1)$ to $\mathcal{D}_k$
10:     **end for**
11:     $\pi_{\theta^*} \leftarrow \text{FINETUNE}(\pi_\theta, \mathcal{D}_k)$
12: **end for**

---

### 4.3 Refinement Ranking and Selection Methods

To select the best refinements to train $\pi_\theta$ on, we compare the following methods.

**Generating Refinements** We condition FeedME on the initial summaries of our train dataset (generated with FeedME) and the human-written feedback and generate 5 refinements $x_1^1, ..., x_1^5$ using the instructions in App. B.1.

**Scoring Refinements with InstructRM** We chose a refinement with a scoring function $R$ that scores refinements for how effectively they incorporate feedback. For $R$ we use the instruction-fine-tuned LM FeedME and ask it whether a refinement is better than the initial summary (see §4.1 for more details). We then evaluate the probability that the refinement incorporates language feedback on the initial summary and is accordingly a high-quality summary, i.e., $p(y_{\text{good}}|\text{prompt})$. LMs are sensitive to the exact prompt used (Perez et al., 2021; Lu et al., 2021), so we write 5 different prompts (see App. B.2) and select the refinement with the highest average $p(y_{\text{good}}|\text{prompt})$ and call this method InstructRM Ensemble.

**Scoring Refinements with Embedding Similarity** Previous work (Scheurer et al., 2022) use a contrastive pre-trained text-embedding function (Neelakantan et al., 2022) to embed the feedback $f$ and refinements $x_1^1, ..., x_1^5$ and select the refinement with the highest cosine similarity to the feedback. They use this scoring function because feedback would often describe what the ideal text should look like. This method is less general because it assumes that good refinements are semantically similar to the feedback, which is not necessarily the case for all tasks or forms of feedback.

#### 4.3.1 Refinement Ranking Results

We now evaluate the above ranking methods on the development dataset by calculating the fraction of times the refinement selected by a method is better than a randomly-selected refinement ("win rate"), according to a ranking given by human evaluators (see App. B.5 for more details). The results, shown in Table 5, show that the embedding similarity selection does not outperform random selection, while most (4/5) InstructRM

Table 5: We compare various ranking methods for selecting refinements using a human evaluation. InstructRM Ensemble is used throughout our paper.

|  | Scoring Function | Win Rate in % vs. Random Selection |
|---|---|---|
| Task-Specific Heuristic | Max Length | $65.0 \pm 2.7$ |
| Zero-Shot | Embedding Similarity | $48.3 \pm 3.0$ |
|  | InstructRM Prompt 1 | $55.0 \pm 3.0$ |
|  | InstructRM Prompt 2 | $58.0 \pm 2.9$ |
|  | InstructRM Prompt 3 | $56.5 \pm 2.9$ |
|  | InstructRM Prompt 4 | $55.8 \pm 2.8$ |
|  | InstructRM Prompt 5 | $50.0 \pm 3.0$ |
|  | InstructRM Ensemble | $56.0 \pm 3.0$ |

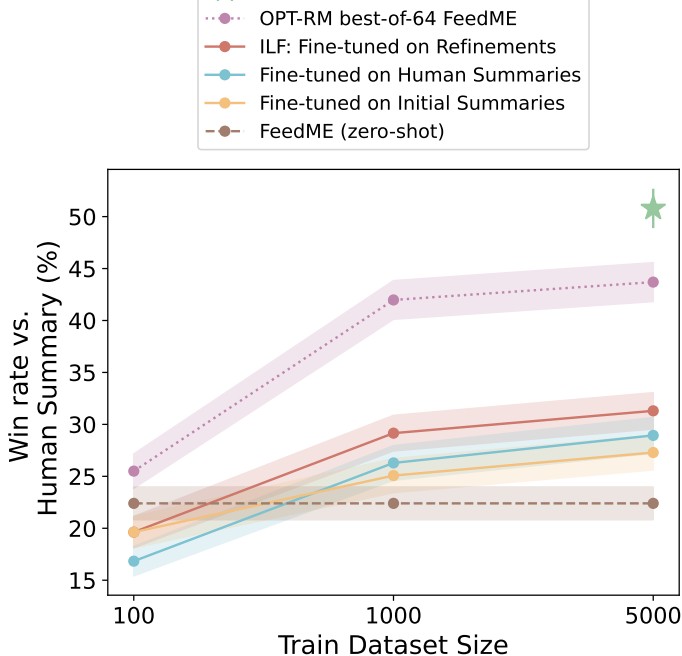

Figure 6: The rate with which human evaluators prefer summaries generated by all methods. The solid lines mark the fine-tuning methods and the dashed brown line marks zero shot generation. "ILF + OPT-RM (best-of-64)" combines learning from natural language feedback and binary preferences, and "OPT-RM best-of-64 FeedME" uses learning from binary preferences to rank and select the best summaries generated by FeedME. "ILF + OPT-RM (best-of-64)" generates summaries of a similar quality to human summaries.

prompts do. While the embedding similarity worked well in previous work (Scheurer et al., 2022), it does not perform well on our dataset. We believe this is because the feedback we collect, written by many annotators, is much more diverse, while in Scheurer et al. (2022), the authors wrote the feedback themselves. InstructRM Ensemble has a win rate of $56.0 \pm 3.0\%$ against random selection, demonstrating that an LM can evaluate its own output to some extent. Based on these results, we recommend using the InstructRM Ensemble approach, as it performs well and is less sensitive to the particular prompt.

Throughout our paper, we use InstructRM Ensemble as our scoring function to select refinements and refer to our method of generating and selecting refinements as *Refinement with Feedback + Best of N*.

### 4.4 Comparing Feedback Learning Algorithms

In this section, we compare ILF against other methods for learning from feedback, including learning from binary feedback and supervised fine-tuning. We present an overview of each method and then provide the results of our evaluations.

#### 4.4.1 Feedback Learning Methods

**Fine-tuning on Refinements (ILF)** For this evaluation, we use a single iteration of ILF to learn from language feedback. We fine-tune GPT3-175B (davinci) (Brown et al., 2020)[8] to maximize the log-likelihood of the refinement given the input prompt (consisting of the Reddit title, and post), i.e., $\log p(x_1|\text{prompt})$, using the refinements generated with Refinement with Feedback + Best of N. For all our fine-tuning methods we add $\lambda \log p(\text{prompt})$ to the loss (Radford & Narasimhan, 2018; OpenAI, 2022a), which maximizes the log-probability of the prompt. The prompt-loss weight $\lambda \in [0,1]$ is chosen on our development dataset (see paragraph "Fine-tuning on Human Summaries"). The selected hyperparameters are detailed in App. C.4 and the fine-tuning prompts in App. B.3.

**Fine-tuning on Human Summaries** Here we fine-tune GPT3-175B on the dataset of human-written summaries $x_{\text{human}}$, with the objective of maximizing the log-probability of human summaries given the input prompt (consisting of the Reddit title and post) with the additional loss term, i.e. $\log p(x_{\text{human}}|\text{prompt}) + \lambda \log p(\text{prompt})$. To ensure the best performance of our fine-tuned models, we conduct thorough hyperparameter tuning on the human-written summary datasets of various sizes (100, 1K, 5K). The hyperparameters optimized include the number of training epochs, the prompt loss weight $\lambda$, and the learning rate multiplier, as detailed in the OpenAI documentation (OpenAI, 2022a). We use the perplexity of the predicted summaries on the development dataset to select the most effective hyperparameters. The selected hyperparameters are applied to all datasets, i.e., fine-tuning on refinements, initial summaries, and human-written summaries, with the same sample size. More details on hyperparameter tuning can be found in Appendix C.4.

**Fine-tuning on Initial Summaries** We fine-tune GPT3-175B on the dataset of initial summaries (generated by FeedME). The objective is to maximize the log probability of the initial summary given the prompt (consisting of the Reddit title and post) with the additional loss term i.e. $\log p(x_0|\text{prompt}) + \lambda \log p(\text{prompt})$. Details on hyperparameter tuning can be found in the paragraph *Fine-tuning on Human Summaries* and Appendix C.4.

**Learning from Binary Feedback: Best-of-$N$** We compare ILF against binary feedback as a baseline, the standard approach for learning from feedback. One way of learning from binary feedback is to train a reward model and use it to do best-of-$N$ sampling. We use best-of-N because it is often competitive with RL from human feedback (Nakano et al., 2021), a highly effective but more sophisticated approach Stiennon et al. (2020); Ouyang et al. (2022). To train the RM, we fine-tune OPT-13B (OPT-RM) (Zhang et al., 2022) to classify whether a summary $x_0$ is high quality or not. To do so, we use the instruction *Is the above an excellent summary of the given text? An excellent summary is coherent, accurate, concise, and detailed. Answer with Yes or No.*, where the label $y$ is either $y_{\text{good}}$ (" Yes") or $y_{\text{bad}}$ (" No"). Given human labels on which of two summaries is preferred, we label the preferred summary with $y_{\text{good}}$ and the other summary with $y_{\text{bad}}$. We then fine-tune the LM to maximize $\log p(y|x_0) + \lambda \log p(x_0)$, where $\lambda \in [0,1]$, chosen using the development dataset, and $y \in \{y_{\text{good}}, y_{\text{bad}}\}$. Using the fine-tuned LM, we evaluate a given summary by computing $p(y_{\text{good}}|x_0)$ and select the summary with the higher probability. We find that this approach leads to more accurate RMs than other RM training methods, such as the commonly used method from Stiennon et al. (2020); see Appendix C.3 for comparisons and Appendix B.4 for the used prompts. We perform Bayesian hyperparameter optimization for OPT-RM and sweep over the learning rate, batch size, and prompt-loss weight $\lambda$, using classification accuracy on the development dataset as the selection criteria (see Appendix C.4 for more details).

---

[8]FeedME cannot be fine-tuned via OpenAI's API.

**ILF + Learning from Binary Feedback**   As a final step, we combine ILF and learning from binary feedback, by first fine-tuning GPT3-175B on the refinements as described in the paragraph fine-tuning on refinements (ILF). We then train the reward model, OPT-RM, and use it to perform best-of-$N$ sampling, as outlined in the paragraph on learning from binary feedback. At test time, we generate 64 summaries with our fine-tuned model and rank them based on their probability of being a high-quality summary, $p_{\mathrm{norm}}(y_{\mathrm{good}}|x_0)$, using OPT-RM. The summary with the highest normalized probability is then selected.

### 4.4.2   Evaluation of Feedback Learning Methods

We evaluate the effectiveness of our learning algorithm, by comparing it to human written reference summaries, several fine-tuning baselines, and OPT-RM on the task of text summarization using 100, 1K, and 5K train samples. Using a test dataset of 698 samples, we generate a summary for each method and evaluate them with human evaluators who rank them based on quality, using a standard ranking scheme that allows for ties between summaries (see App. C.4 for more details). Based on the rankings, we calculate the fraction of times each method's sampled summary outperforms the human-written reference summary, referred to as the "win rate". We sample summaries up to 48 tokens in length (as in Stiennon et al. (2020)) using nucleus sampling (Holtzman et al., 2019) with $p = 0.95$ and temperature $t = 1.0$ (see App. C.4 for further details on hyperparameters and postprocessing). We use best-of-64 sampling with summaries sampled from FeedME for learning from binary feedback.

### 4.4.3   Results

**ILF Versus Other Fine-Tuning Methods**   Our results, shown in Fig. 6, demonstrate that fine-tuning on refinements (ILF) outperforms all other fine-tuning methods[9]), including sampling from FeedME, with a win rate against human summaries of 31.3±1.7% (for fine-tuning on 5K samples), while the other methods achieve win rates of $27.3 \pm 1.7\%$ (fine-tuning on initial summaries), $28.9 \pm 1.7\%$ (fine-tuning on human summaries), and $22.5 \pm 1.6\%$ (FeedME). It is surprising that ILF outperforms fine-tuning on human summaries across all sample sizes, despite human-written summaries generally being of higher quality (see Fig. 15, top). Further evaluation (see App. C.8, Fig. 11) shows that the model fine-tuned on 1K refinements (ILF) exhibits significantly lower loss when evaluated on the validation dataset of refinements compared to the model fine-tuned on human summaries when evaluated on the validation dataset of human summaries, suggesting that the model is more adept at approximating the distribution of refinements. Overall, these results demonstrate the effectiveness of our proposed ILF approach in accurately incorporating feedback and improving model performance, even outperforming fine-tuning on human summaries.

(Scheurer et al., 2022) found that ILF with 100 feedback samples outperformed FeedME, while here we find it underperforms FeedME with 100 feedback samples. Prior work uses author-written feedback that often conveys what the refinement should include, while our work includes more varied, crowdsourced feedback. As a result, we observe that embedding similarity does not properly rank refinements on our human feedback dataset (Table 5), and we believe the difference in feedback may be a significant source of differences in results in this section as well; see Appendix C.8.4 for more discussion.

**Combining ILF With Learning From Binary Feedback**   Our results demonstrate that using OPT-RM for best-of-64 sampling on FeedME summaries outperforms all fine-tuning methods and sampling approaches across all sample sizes. The improved performance of OPT-RM best-of-64 FeedME comes at the cost of added inference time for best-of-$N$ sampling. Combining ILF and learning from binary feedback (ILF + OPT-RM (best-of-64)) achieves human-level summarization performance with a win rate of $50.8 \pm 1.9\%$ using 5K samples for training. This suggests that both methods independently learn valuable information about human preferences that can be cumulative when used together. It should be noted that the result for ILF + OPT-RM (best-of-64) is obtained through a separate human evaluation with different comparison summaries (see App. Fig. 12), and was added to Fig. 6 for reference.

---

[9]Fine-tuning on 100 refinements is tied with fine-tuning on 100 initial summaries.

Lastly, we provide further analyses in Appendixes C.8.5 and C.9 demonstrating that the feedback is important for high-quality refinements and that the refinements do frequently incorporate the most important point in the feedback.

## 5 Related Work

**Learning from Human Feedback**   Our algorithm is inspired by a number of past works that have trained models to learn from feedback. A common technique is reinforcement learning from human feedback (RLHF Christiano et al., 2017; Ziegler et al., 2019; Stiennon et al., 2020; Ouyang et al., 2022), which trains models to satisfy human preferences. However, our algorithm is closer to works that use natural language feedback, rather than comparisons between different choices. Elgohary et al. (2020); Austin et al. (2021); Nijkamp et al. (2022) all demonstrate that code LLM performance generally improves when prompted with natural language feedback, though Nijkamp et al. (2022) observes that the feedback is more effective when it is given one step at a time. Our work differs from these in that ILF learns from the feedback at training time, not at inference time.

Bai et al. (2022a) also uses natural language feedback during the training process, but as part of an RLHF algorithm instead where the feedback is used to solicit different responses from the digital assistant, the responses are ranked by crowdworkers, and the rankings are used to train the preference model. However, they note that this form of learning from natural language feedback does not measurably improve their code generation model more than simply prompting.

Other work aims to use natural language as a learning signal in other ways. Radford et al. (2019), Raffel et al. (2020), and Brown et al. (2020) investigate using explanations for *gold labeled outputs* of classification tasks, whereas our work addresses the more general text generation setting. Furthermore, explanations are notably different from feedback – explanations describe why a labeled output is correct, whereas feedback describes how to improve a candidate output. Explanations can also be used to train text classification models, with mixed results (Camburu et al., 2018; Stacey et al., 2021; Pruthi et al., 2021; Wiegreffe et al., 2021; Hase & Bansal, 2021; Lampinen et al., 2022, *inter alia*). In addition, some work learns from language feedback for ranking rather than generating outputs (Weston, 2016; Li et al., 2016; Hancock et al., 2019; Li et al., 2022b; Xu et al., 2022b). Matiana et al. (2021) learn text embeddings of language feedback, where improvements could benefit the refinement-scoring step of our algorithm.

Language has also been widely used in RL settings for various purposes (see Luketina et al., 2019, for an overview), such as specifying tasks ("instruction following"; Wei et al., 2022a; Ouyang et al., 2022, *inter alia*), driving exploration (Tam et al., 2022), inferring reward functions (Lin et al., 2022; Sumers et al., 2021; Fidler et al., 2017, *inter alia*), and training a model via strong supervision Andreas et al. (2017); Kaplan et al. (2017), reward shaping Goyal et al. (2019), or by providing descriptions of trajectories (Nguyen et al., 2021). In contrast, we use language to correct faulty behavior. Other work uses language feedback at test time to correct mistakes in a model's behavior, e.g., image segmentation (Rupprecht et al., 2018) or code generation Elgohary et al. (2020); Austin et al. (2021). In contrast, we use feedback to *train* models, and our approach does not require human intervention at test time.

Lastly, our work builds upon our previous report (Scheurer et al., 2022), which showed that LLMs can refine outputs with language feedback. There, we introduced the same three-step algorithm that ILF builds upon and applied the algorithm only to text summarization. Our present work extends (Scheurer et al., 2022) in a number of ways, including: (1) formalizing the theoretical justification of ILF (Section 2); (2) adapting ILF to multiple tasks (*e.g.* both code generation and text summarization); (3) conducting more extensive summarization experiments analyzing the best methods for ranking refinements and selecting the best final summary; and (4) using larger-scale human crowdsourcing to both rate output summaries and generate feedback and refinements.

**LLMs for Program Synthesis**   Our work also builds on a large body of literature that explores the use of pre-trained LLMs for neural program synthesis. Many general purpose LLMs, although not pre-trained specifically for code generation, have demonstrated impressive proficiency at solving code challenges since they are pre-trained on large corpora of text such as THE PILE (Gao et al., 2020) that contain a small

percentage of code content (Austin et al., 2021; Wang & Komatsuzaki, 2021; Black et al., 2022; Nijkamp et al., 2022). Yet other recent LLMs for program synthesis are trained on solely source code files (Wang et al., 2021; Zan et al., 2022; Li et al., 2022a; Xu et al., 2022a), or on both text and source code documents – sometimes either in succession (Chen et al., 2021; Nijkamp et al., 2022; Bai et al., 2022a), in a mixed corpus (Workshop et al., 2022), or on mixed natural language-programming language documents (Feng et al., 2020).

**Bayesian Inference for LLMs**  Several other works draw connections between Bayesian Inference and learning algorithms for LMs. Korbak et al. (2022) show that KL-regularised RL is equivalent to variational inference: approximating a Bayesian posterior which specifies how to update a prior LM to conform with evidence provided by a reward function. Dohan et al. (2022) further argues that the process of generating output through multiple rounds of interaction between prompted LMs and other agents (e.g. humans providing language feedback) can be seen as executing probabilistic programs.

## 6  Conclusion

We have shown that ILF can significantly improve the quality of LLM generations for both code generation and text summarization, even with just a small sample of human-written feedback and refinements. Combining ILF with learning from binary preferences further improves the quality of the LLM's outputs. We have also shown that this approach is theoretically justified as minimizing the expected KL divergence between $\pi_\theta$ and a target ground-truth distribution, where we acquire signal from the latter via human-written natural language feedback.

This approach is also appealing because it is not model-specific (in the sense that ILF can be used with any type of base model $\pi_\theta$, assuming the existence of a sufficiently capable LLM to act as $\pi_{\text{Refine}}$), and can be conducted in multiple rounds to continuously improve the model. ILF also does not involve training separate policies for modeling the reward signal and advantage values, as RL-based feedback learning algorithms often do (Stiennon et al., 2020). Furthermore, it is notable that our approach generates training data that is not only correct, but targets the specific kinds of weaknesses that are more likely in the given model's generations. In essence, it provides an *online* and *on-policy* training signal that is missing from the offline pre-training and fine-tuning set-ups of modern LLMs. Our approach is also remarkably sample-efficient, improving the pass@1 rate from 22% to 36% compared to the baseline, despite fine-tuning on only 78 examples. For summarization, we train on only 5000 examples to achieve approximately human-level summarization quality.

Our work opens up multiple avenues for promising future work. For instance, ILF can be applied iteratively over the course of multiple rounds whenever new information arrives (*e.g.* new Python syntax or world knowledge) or new bugs/quality issues are discovered. Our preliminary experiments in Appendix C.8.2 indicate that ILF benefits from the data gathered over multiple rounds, although continual fine-tuning may cause the model to occasionally forget past feedback. More extensive experimentation is required to study how to effectively apply numerous rounds of ILF. Additionally, the tasks we study in this paper require only shorter outputs, and it remains to be seen how ILF generalizes to longer output sequences. It would also be useful to study how to generalize the implementation of ILF across a variety of domains and tasks. In this work, we designed separate reward functions for each task, but it would be beneficial to explore more generalizable reward functions that are easier to transfer between domains.

As the pace of progress of modern LLM research continues to accelerate, it may soon be feasible to partially or fully automate the generation of natural language feedback (similar to 'supervised learning from constitutional AI' (SL-CAI; Bai et al., 2022b) and our experiments in Section 3.4), greatly reducing both the time and cost necessary for collecting feedback. This direction of work is also particularly appealing because the learning signal is *process-based* rather than outcome-based, which has been shown to mitigate reward hacking and improve the correctness of intermediate reasoning steps (Uesato et al., 2022).

Although further work is required to extend our method, ILF represents an exciting step forward in training LLMs with feedback that is rich, interactive, and sample-efficient.

## 7 Acknowledgements

We are grateful to Nitarshan Rajkumar, Jason Phang, Nat McAleese, Geoffrey Irving, Jeff Wu, Jan Leike, Cathy Yeh, William Saunders, Jonathan Ward, Daniel Ziegler, Seraphina Nix, Quintin Pope, Kay Kozaronek, Peter Hase, Talia Ringer, Asa Cooper Stickland, Jacob Pfau, David Lindner, Lennart Heim, Kath Lumpante, and Pablo Morena for helpful discussions and feedback about the design and implementation of this work. We are additionally thankful to Scott Heiner and Edwin Chen for extensive help with setting up our human annotation workflow and interface. We also thank the TMLR reviewers who provided valuable feedback that meaningfully improved this paper.

EP thanks the National Science Foundation and Open Philanthropy for fellowship support. JAC is supported by a doctoral grant from the Spanish MECD. AC, SB, and KC are supported by National Science Foundation Awards 1922658 and 2046556. Any opinions, findings, and conclusions or recommendations expressed in this material are those of the author(s) and do not necessarily reflect the views of the National Science Foundation. KC is additionally supported by 42dot, Hyundai Motor Company (under the project Uncertainty in Neural Sequence Modeling) and the Samsung Advanced Institute of Technology (under the project Next Generation Deep Learning: From Pattern Recognition to AI). This project has also benefited from financial support to SB by Eric and Wendy Schmidt (made by recommendation of the Schmidt Futures program), Open Philanthropy, and Apple. We also thank the NYU High-Performance Computing Center for in-kind support and OpenAI for providing access to and credits for their models via the API Academic Access Program.

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

# A   Appendix

## A.1   Code Generation Prompt Formats

<table>
<tr><td>Prompt</td><td>Expected completion</td></tr>
<tr><td>

```
OLD CODE:
"""
Write a python function to find
the sum of the three lowest
positive numbers from a given list
of numbers.
>>> Example:
sum_three_smallest_nums([10,20,30,
40,50,60,7]) = 37
"""
def sum_three_smallest_nums(lst):
    lst.sort()
    return sum(lst[:3])

FEEDBACK:
This code finds the sum of the smallest 3
numbers, not the smallest 3 positive numbers.
It needs to disregard negatives and 0.

REFINEMENT:
```

</td><td>

```
"""
Write a python function to find
the sum of the three lowest
positive numbers from a given list
of numbers.
>>> Example:
sum_three_smallest_nums([10,20,30,
40,50,60,7]) = 37
"""
def sum_three_smallest_nums(lst):
    lst = [x for x in lst if x >
0]
    lst.sort()
    return sum(lst[:3])
```

</td></tr>
</table>

Figure 7: An example of a zero-shot LLM prompt for repairing incorrect code based on human-written feedback.

## A.2   Code Generation Training Details

For the experiments in Section 3.3, we run a hyperparameter sweep for all methods except for ILF. The hyperparameter value ranges that we sweep include learning rate $\in \{1.0^{-6}, 5.0^{-6}, 1.0^{-5}\}$, batch size $\in \{32, 64, 128\}$, and number of epochs $\in \{1, 2, 5\}$. The tasks for the training and validation datasets are from $\mathrm{MBPP}_{\mathrm{Train}}$ and $\mathrm{MBPP}_{\mathrm{Refine}}$, respectively, while the programs are sourced from the method (*e.g.* InstructGPT, MBPP, human-written, or zero-shot CODEGEN-MONO 6.1B). For ILF, we use the best hyperparameters obtained for the sweep over MBPP programs instead of sweeping over ILF-generated programs, since the tasks in $\mathrm{MBPP}_{\mathrm{Refine}}$ are already used to train $\pi_{\mathrm{Refine}}$. All pass rates reported in Table 4 are obtained by evaluating each method on $\mathrm{MBPP}_{\mathrm{Test}}$ using the best hyperparameters found during the sweep on $\mathrm{MBPP}_{\mathrm{Refine}}$.

For the experiments in Section 3.4, we separately tune hyperparameters for each size of dataset. As in our other experiments, we train and validate using the tasks from $\mathrm{MBPP}_{\mathrm{Train}}$ and $\mathrm{MBPP}_{\mathrm{Refine}}$, respectively, coupled with the refinements generated by InstructGPT that pass the unit test suites. We sweep the same hyperparameter value ranges as the experiments in the previous section (*i.e.* learning rate $\in \{1.0^{-6}, 5.0^{-6}, 1.0^{-5}\}$, batch size $\in \{32, 64, 128\}$, and number of epochs $\in \{1, 2, 5\}$).

We implement all experimental pipelines with the HuggingFace transformers (v4.12.5) (Wolf et al., 2020), Huggingface datasets (v2.7.1) (Lhoest et al., 2021), and Pytorch (v1.11) (Paszke et al., 2019) libraries.

### A.3 Code Generation Annotator Instructions

Given a natural language description of a Python programming challenge and its accompanying unit tests, you will be shown **10 sample model-generated Python solutions that do not pass the tests**. Please do the following:

1) Select one model-generated code sample that seems relatively easy to correct (such that it can be minimally corrected to pass the unit tests). If no such code sample exists (ie every code sample would require extensive correction, select the corresponding option and move on to the next task.

2) Write 1-4 sentences of **natural language feedback** for the code sample that does two things: **(a) describes what is wrong with the code sample,** and **(b) how it can be fixed.** You can use individual variable or method names, but please **do not include entire lines of code**. Try to describe the necessary logic using mostly natural language, not Python expressions. Below are some examples of good versus bad feedback:

**Good:**

> "The current code is wrong because it returns items using the heappop method in the hq module, which gets the smallest items even though the task description actually asks for the largest items. Instead, the code should use the nlargest function in hq to get the largest n integers."

**Bad (because it only describes what to change, and not what was originally wrong with the code):**

> "The code should use the nlargest function in hq to get the largest n integers."

**Bad (because it gives the actual code needed to fix the function, instead of describing the desired fix using words only):**

> "The current code is wrong because it returns items using the heappop method in the hq module, which gets the smallest items even though the task description actually asks for the largest items. Instead, the function should return [hq.heappop(nums) for i in range(n)]."

3) Write a **refinement** for the code that consists of copying the original model sample and making minimal edits to the code such that it correctly solves the challenge and can pass all the unit tests.

Here is an example. We only show a couple of model samples here for brevity, but in practice you will be shown up to 10 model samples per task.

**Example 1**:

Prompt:

```
"""

Write a function to find the similar elements from the given two tuple lists.

>>> Example: similar_elements((3, 4, 5, 6),(5, 7, 4, 10)) = (4, 5)

"""

def similar_elements(test_tup1, test_tup2):
```

Unit Tests:

```
def check ( candidate ):

        assert candidate ((3, 4, 5, 6),(5, 7, 4, 10)) == (4, 5), "candidate ((3, 4, 5,
    6),(5, 7, 4, 10)) == (4, 5)"

        assert candidate ((1, 2, 3, 4),(5, 4, 3, 7)) == (3, 4), "candidate ((1, 2, 3, 4)
    ,(5, 4, 3, 7)) == (3, 4)"

        assert candidate ((11, 12, 14, 13),(17, 15, 14, 13)) == (13, 14), "candidate
    ((11, 12, 14, 13),(17, 15, 14, 13)) == (13, 14)"
```

Model Sample 0:

```
"""

Write a function to find the similar elements from the given two tuple lists.

>>> Example: similar_elements ((3, 4, 5, 6),(5, 7, 4, 10)) = (4, 5)

"""

def similar_elements ( test_tup1 , test_tup2 ):

    """

    Similar to sets , I want to get the values that are in both the given lists.

    """

    if len ( test_tup1 ) == len ( test_tup2 ):

        return set ( test_tup1 ) & set ( test_tup2 )

    else :

        raise Exception ("Tuple lists of different lengths ")
```

*Unit test results for model sample 0:* `failed:  candidate ((3, 4, 5, 6),(5, 7, 4, 10)) == (4, 5)`

Model Sample 1:

```
"""

Write a function to find the similar elements from the given two tuple lists.

>>> Example: similar_elements ((3, 4, 5, 6),(5, 7, 4, 10)) = (4, 5)

"""

def similar_elements ( test_tup1 , test_tup2 ):

    """

    Args :

    test_tup1 and test_tup2 are two list of tuples of same length

    Returns :

    a tuple of similar elements from the first list and second list
```

```
    """

    res = []

    for x,y in zip(test_tup1, test_tup2):

        res.append(x if x == y else (x,y))

    return tuple(res)
```

*Unit test results for model sample 1*: `failed:  candidate((3, 4, 5, 6),(5, 7, 4, 10)) == (4, 5)`

Example annotator response:

1) Which model completion are you choosing to write feedback and a refinement for?

*Completion 0*

2) What feedback do you have for this model sample? Write 1-4 sentences of **natural language feedback** for the code sample that does two things: **(a) describes what is wrong with the code sample,** and **(b) how it can be fixed**. You can use individual variable or method names, but please **do not include entire lines of code**. Try to describe the necessary logic using mostly natural language, not Python expressions.

*The logic is mostly correct, but the function returns the wrong type. It should always return a tuple, not a set or Exception. The return statement should convert the output to a tuple prior to returning, and the raise Exception statement should return an empty tuple instead.*

3) How can you ***minimally edit*** the code such that it correctly solves the challenge and can pass all the unit tests? (Please copy over the original completion and edit it **only as necessary.**)

```
    """

    Write a function to find the similar elements from the given two tuple lists.

    >>> Example: similar_elements((3, 4, 5, 6),(5, 7, 4, 10)) = (4, 5)

    """

    def similar_elements(test_tup1, test_tup2):

        """

        Similar to sets, I want to get the values that are in both the given lists.

        """

        if len(test_tup1) == len(test_tup2):

            return tuple(set(test_tup1) & set(test_tup2))

        else:

            return tuple()
```

### A.4 Alternative $\pi_{\text{Refine}}$ Models

We also explore prompting pre-trained LLMs such as `gpt-3.5-turbo` and `gpt4` as $\pi_{\text{Refine}}$, instead of fine-tuning a CODEGEN-MONO 6.1B model. Although such models are closed-source and require API credits to access, the use of few-shot prompting removes the need to acquire a separate training dataset and computational resources for fine-tuning a custom $\pi_{\text{Refine}}$ model. We use the same prompt format as in Appendix A.1, but with two in-context examples added. The results are shown in Table 6. Although the custom-trained $\pi_{\text{Refine}}$ model still yields the strongest results, both `gpt-3.5-turbo` and `gpt-4` still yield significantly higher pass@1 and pass@10 rates compared to both the zero-shot baseline and ablations in Table 4. As expected, all the variations on ILF perform comparably to the gold standard of training on human-written refinements (the second-to-last row in Table 4). This demonstrates that ILF effectively improves the base model even when different kinds of $\pi_{\text{Refine}}$ model are used, and does not necessarily require extra data for training $\pi_{\text{Refine}}$.

Table 6: Final performance of $\pi_{\theta*}$ (trained with ILF) on MBPP$_{\text{Test}}$, compared across multiple types of $\pi_{\text{Refine}}$ models.

| Method | Feedback Source | Source of Refinements | Pass Rates of $\pi_{\theta*}$ | |
| --- | --- | --- | --- | --- |
| | | | Pass@1 | Pass@10 |
| | Humans | Fine-tuned CODEGEN-MONO 6.1B | **36%** | **68%** |
| ILF | Humans | 2-shot `gpt-3.5-turbo` | 34% | **68%** |
| | Humans | 2-shot `gpt-4` | 32% | 65% |

### A.5 Human Feedback Is More Informative Than InstructGPT Feedback

Table 7: The proportion of the feedback that addressed each type of bug, for feedback sourced from humans and InstructGPT. Each sample of feedback can be tagged with multiple categories, so the quantities in each column do not necessarily add up to 100%.

| Feedback Category | % of Feedback | |
| --- | --- | --- |
| | Human | InstructGPT |
| Logic | 30% | 46% |
| Formatting | 36% | 14% |
| Missing step | 10% | 6% |
| Algebra | 10% | 8% |
| Recursion | 4% | 14% |
| Regex | 6% | 6% |
| Function semantics | 2% | 4% |
| Dynamic programming | 2% | 0% |
| Extra step | 0% | 12% |
| No feedback needed | 0% | 14% |
| Unrelated | 0% | 8% |

To better understand why human feedback produced greater improvements in pass rate than InstructGPT feedback, we randomly selected 50 samples of feedback for each source (*i.e.* human or InstructGPT) and annotated the number and types of bugs that each feedback sample addressed. The results are shown in Tables 7 and 8. We observed that InstructGPT often gave no feedback (*e.g.* "The code is correct" or "Great job!"), provided feedback that was irrelevant or incorrect, or restated the task description instead of addressing what should be repaired about the code. Despite this, InstructGPT's refinements were often correct even if the feedback itself wasn't. Human-written feedback addressed more bugs on average and did not contain irrelevant feedback. We provide further examples of the differences between human and InstructGPT feedback in Appendix A.7.

Table 8: Descriptive statistics for the human- versus InstructGPT-generated feedback. The * indicates that the metric was computed on the random sample of 50 that we manually inspected, whereas the other metrics are computed from the full dataset.

|  | Source of Feedback | |
|---|---|---|
|  | Human | Instruct-GPT |
| Avg. num. of bugs addressed* | 1.8 | 1.1 |
| Avg. num. of words | $68.9 \pm 48.2$ | $24.2 \pm 28.6$ |

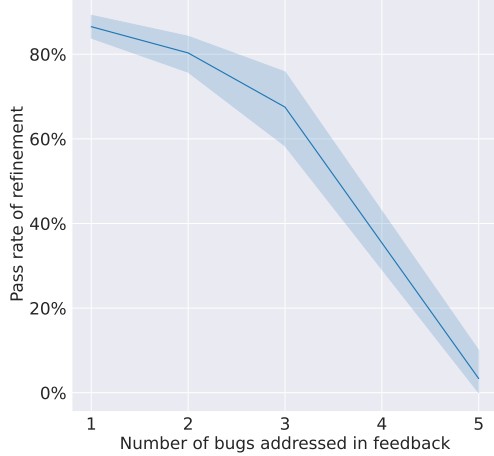

Figure 8: The number of bugs addressed in the feedback versus the pass rate of $\pi_{\text{Refine}}$'s refinements.

### A.6    $\pi_{\text{Refine}}$ Struggles To Incorporate Feedback Addressing Many Bugs

Lastly, we explored whether the number of bugs addressed in the feedback affected $\pi_{\text{Refine}}$'s ability to repair the original code sample. The results are shown in Figure 8. The greater the number of bugs addressed, the lower the average pass rate of $\pi_{\text{Refine}}$'s refinements. This suggests that a promising direction for future work might consist of automatically decomposing the feedback into multiple steps and having $\pi_{\text{Refine}}$ incorporate the feedback one step at a time. Indeed, Nijkamp et al. (2022) show that the CODEGEN models are often more effective at following instructions when the instructions are given across multiple turns, and recent Chain-of-Thought work (Wei et al., 2022b) illustrates a similar prompting technique.

### A.7 Examples of Human Versus InstructGPT Feedback on Generated Programs

Table 9: Examples of natural language feedback written by humans versus InstructGPT for incorrect programs generated by CODEGEN-MONO 61.B.

| Incorrect program generated by CODEGEN-MONO 6.1B | Human feedback | InstructGPT feedback |
|---|---|---|
| ```python """ Write a function to assign     frequency to each tuple in     the given tuple list.  >>> Example: assign_freq([(6,     5, 8), (2, 7), (6, 5, 8),     (6, 5, 8), (9, ), (2, 7)] )      = '[(6, 5, 8, 3), (2, 7,     2), (9, 1)]' """  from collections import     Counter def assign_freq(test_list):         """         >>> assign_freq([(6,     5, 8), (2, 7), (6, 5, 8),     (6, 5, 8), (9, ), (2, 7)] )      = '[(6, 5, 8, 3), (2, 7,     2), (9, 1)]'         """         counter = Counter(x     for x in test_list)         res = []         for key, val in     counter.items():                 temp = (key,     val)                 res.append(     temp)         return str(res) ``` | "The only issue with this code is that it does not dereference the key. When you set temp = (key, val), the key is an entire tuple instead of the values in the tuple. Thus, you need the asterisks to get the values" | "The function works as intended." |
| ```python """ Write a function to get the     angle of a complex number.  >>> Example: angle_complex(0,1     j)=1.5707963267948966 """  import cmath def angle_complex(a,b):         return cmath.phase(     complex(a,b)) ``` | "Only one thing is wrong in the code that complex method can take the complex number as argument example complex(a+b) not like complex(a,b) we have to replace , by + and the code is fine." | "The code should use the atan2 function to get the angle of a complex number." |

Table 9: Continued from previous page

| Incorrect program generated by CODEGEN-MONO 6.1B | Human feedback | InstructGPT feedback |
|---|---|---|
| ```python
"""
Write a function to count the
    same pair in three given
    lists.

>>> Example: count_samepair
    ([1,2,3,4,5,6,7,8],
    [2,2,3,1,2,6,7,9],
    [2,1,3,1,2,6,7,9])=3
"""

def count_samepair(list1,list2
    ,list3):
    count = 0
    for i in range(len(list1))
    :
        for j in range(len(
    list2)):
            if list1[i]==list2
    [j] and list2[j]==list3[i]:
                count+=1
    return count
``` | "One extra for loop for list2 is causing the issue, it finds more matching elements between list1,2,3. The possible solution for the code is to remove the unnecessary for loop of list2 iteration, and check all the lists same index element at the time, and return the count of this element." | "The code above is very inefficient. It is checking every possible combination of pairs of elements in the three lists. It would be much more efficient to use a set to keep track of the pairs that have already been seen." |
| ```python
"""
Write a function to find
    number of lists present in
    the given tuple.

>>> Example: find_lists(([1,
    2, 3, 4], [5, 6, 7, 8])) =
    2
"""

def find_lists(Input):
    result = 0
    for x in Input:
        if isinstance(x, list)
    :
            result += 1
    return result
``` | "The code works for tuples of more than one list, but not in case of one single list. Python goes one level down and reads the elements when given a single list and does not see an instance of list. I would just check for the case of 'type(Input) == list' and turn the input into a 1-sized list of lists in this case." | "The function find_lists needs to return the number of lists present in the given tuple, not the number of items in the tuple." |

Table 9: Continued from previous page

| Incorrect program generated by CODEGEN-MONO 6.1B | Human feedback | InstructGPT feedback |
|---|---|---|
| """
Write a function to calculate
    magic square.

>>> Example: magic_square_test
    ([[7, 12, 1, 14], [2, 13,
    8, 11], [16, 3, 10, 5], [9,
     6, 15, 4]])=True
"""

def magic_square_test(
    my_matrix):
    """
    >>> magic_square_test([[7,
     12, 1, 14], [2, 13, 8,
    11], [16, 3, 10, 5], [9, 6,
     15, 4]])=True
    """
    def is_square(x):
        return x==x*x
    def is_magic_square(
    my_matrix):
        sides=[]
        for i in range(len(
    my_matrix)):
            sides.append(sum(
    my_matrix[i]))
        if not is_square(sum(
    sides)):
            return False
        sides=[]
        for i in range(len(
    my_matrix)):
            sides.append(sum([
    my_matrix[i][j] for j in
    range(len(my_matrix))]))
        if not is_square(sum(
    sides)):
            return False
        diagonals=[]
        for i in range(len(
    my_matrix)):
            diagonals.append(
    my_matrix[i][i])
        if not is_square(sum(
    diagonals)):
            return False
        diagonals=[]
        for i in range(len(
    my_matrix)):
            diagonals.append(
    my_matrix[i][len(my_matrix)
    -1-i])
        if not is_square(sum(
    diagonals)):
            return False
        return True
    return is_magic_square(
    my_matrix) | "The logic behind checking whether the sum of rows, columns, and diagonals is a magic number or not is not correct, the code checks if the sum of squares of itself in the is_square method is wrong. Instead of checking all the sums of rows, columns, and diagonals at every point, the logic should be like saving all the sums for rows, columns, and diagonals in one array(sides) and then checking if there are all same values present in this array because for magic_square_test all the sums should be equal." | "The function magic_square_test should return whether my_matrix is a magic square." |

Table 9: Continued from previous page

| Incorrect program generated by CODEGEN-MONO 6.1B | Human feedback | InstructGPT feedback |
|---|---|---|
| ```python
"""
Write a python function to
    count characters at same
    position in a given string
    (lower and uppercase
    characters) as in english
    alphabet.

>>> Example:
    count_char_position("xbcefg
    ") = 2
"""

def count_char_position(str1):
    counter = 0
    for i in range(0, len(str1
    )):
        if str1[i] == str1[i].
    upper():
            counter += 1
    return counter
``` | "a) The first thing that the code sample gets incorrect is that it increments the counter when each character in the string is an upper case character. The prompt is asking for how many characters in str1 have the same numerical position as their respective alphabetic counterpart and the code does not check for this at all. b) The way to fix it would be to create a separate j variable that keeps track of the ordinal value that corresponds with the alphabetic character that we would expect to be compared to the respective character in str1. j and i would increment at the same time so that the code is both iterating through the alphabet AND iterating through str1 simultaneously. The other fix would be to automatically convert str1 to lowercase so that we don't have to worry about checking the ordinal codes of both the lowercase and uppercase versions of the same alphabetic character." | "Great job!" |

# B ILF for Text Summarization Prompts

## B.1 Summarization Prompts

We report all prompt templates used to generate INTIAL SUMMARIES, REFINEMENT WITH FEEDBACK, and REFINEMENT WITHOUT FEEDBACK in Table 10.

Table 10: Prompt templates used for summarization.

| Methods | Format |
|---|---|
| INITIAL SUMMARY | Write an excellent summary of the given text. |
| | Title: {title} |
| | Text: {text} |
| | TL;DR: |
| REFINEMENT WITH FEEDBACK | Write an excellent summary that incorporates the feedback on the given summary and is better than the given summary. |
| | Title: {title} |
| | Text: {text} |
| | Summary: {summary} |
| | Feedback on Summary: {feedback} |
| | Improved TL;DR: |
| REFINEMENT WITHOUT FEEDBACK | Write an excellent summary that is better than the given summary. |
| | Title: {title} |
| | Text: {text} |
| | Summary: {summary} |
| | Improved TL;DR: |

### B.2 InstructRM Prompts

We instructed one of the authors of this paper (who at the time had not been involved in the research project) to write 5 prompts that would achieve the goal of selecting high-quality summaries, i.e., refinements. The author did not have any domain knowledge or prior information on what kinds of prompts would work. The instructions provided to the author can be viewed here. We report all 5 prompt templates in Table 11.

Table 11: Prompt templates used for InstructRM Ensemble.

| InstructRM Prompts | Format |
|---|---|
| PROMPT 1 | Here's a summary of a Reddit post, feedback on the summary, and a new summary. You will be asked to determine whether the new summary incorporates the feedback provided. |
| | A good summary is a short piece of text that has the essence of the original text. A good summary tries to accomplish the same purpose and conveys the same information as the original text. |
| | Post title: {title} |
| | Below, there's the content of the post that was summarized. |
| | Original post: {text} |
| | Original summary: {summary} |
| | A human then provided feedback on the above summary. |
| | Feedback: {feedback} |
| | Based on this feedback, a new summary was written. |
| | New summary: {refinement} |
| | Does this new summary incorporate the feedback provided? Answer Yes or No. |
| | Answer: |
| PROMPT 2 | Post title: {title} |

Original post: {`text`}

Original summary: {`summary`}

Feedback: {`feedback`}

New summary: {`refinement`}

Question: Does the new summary incorporate the feedback provided? Answer Yes or No.

Answer:

PROMPT 3

You will be given a Reddit post title, its content, an original summary of that post, and feedback for that summary. Then, your goal will be to determine whether the new summary improves upon the original with respect to provided feedback.

Post title: {`title`}

Post content: {`text`}

Original summary: {`summary`}

Feedback: {`feedback`}

New summary: {`refinement`}

Question: Does the new summary incorporate the feedback provided? Answer True or False.

Answer:

PROMPT 4

Here's a summary of a Reddit post, feedback on the summary, and a new summary. You will be asked to determine whether the new summary incorporates the feedback provided.

A good summary is a short piece of text that has the essence of the original text. A good summary tries to accomplish the same purpose and conveys the same information as the original text. Remember, you will be asked to determine whether the new summary incorporates the feedback provided.

Post title: {`title`}

Below, there's the content of the post that was summarized.

Original Post: {`text`}

Remember, you will be asked to determine whether the new summary incorporates the feedback provided. Here's the original summary.

Original summary: {`summary`}

Remember, you will be asked to determine whether the new summary incorporates the feedback provided. A human then provided feedback on the above summary.

Feedback: {`feedback`}

Based on this feedback, a new summary was written.

New summary: {`refinement`}

Does this new summary incorporate the feedback provided? Answer Yes or No.

Answer:

| | |
|---|---|
| PROMPT 5 | Here's a summary of a Reddit post, feedback on the summary, and a new summary. You will be asked to determine whether the new summary incorporates the feedback provided.

The feedback was:

Feedback: feedback |

Here's the post that was summarized in the first place.

Post title: {`title`}

Original Post: {`text`}

Remember, you will be asked to determine whether the new summary incorporates the feedback provided. Here's the original summary.

Original summary: {`summary`}

Remember, you will be asked to determine whether the new summary incorporates the feedback provided. A human then provided feedback on the above summary. Here's the feedback again.

Feedback: {`feedback`}

Based on this feedback, a new summary was written.

New summary: {`refinement`}

Does this new summary incorporate the feedback provided? Answer True or False.

Answer:

## B.3 Fine-tuning Prompts

In Table 12, we report the prompts we use for fine-tuning on summaries and fine-tuning on feedback + refinements. The completion for fine-tuning on summaries indicates that we can have completions generated from various sources, i.e., either initial summaries from FeedME, refinements generated with our method, or ideal human written summaries. For fine-tuning feedback + refinements, we first generate the feedback and then the refinement.

Table 12: Prompt templates used for Fine-tuning on Summaries and Feedback + Refinement.

| Methods | Prompt | Completion |
|---------|--------|------------|

| | | |
|---|---|---|
| FINE-TUNING ON SUMMARIES | Write an excellent summary of the given text.

Title: {`title`}

Text: {`post`}

TL;DR: | {`summary/refinement/human summary`} |
| FINE-TUNING ON FEEDBACK + REFINEMENTS | Write an excellent summary that incorporates the feedback on the given summary and is better than the given summary.

Title: {`title`}

Text: {`post`}

Summary: {`summary`}

Feedback on summary: | {`feedback`}

Improved TL;DR: {`refinement`}
### |

### B.4 Reward Model Prompts

Table 13: Prompt templates used for training the reward model with the language model loss. Both classification and comparison prompts are shown.

| Reward Type | Model | Prompt | Completion |
|---|---|---|---|
| | BINARY RM | Title: {`title`}

Text: {`post`}

TL;DR: {`summary_A/summary_B`}

Question: Is the above an excellent summary of the given text? An excellent summary is coherent, accurate, concise, and detailed. Answer with Yes or No. | {`" Yes"/" No"`} |

| | Answer: | |
|---|---|---|
| COMPARISON RM | Title: {title} | {" A"/" B"} |
| | Text: {post} | |
| | Summary A: {summary_A} | |
| | Summary B: {summary_B} | |
| | Question: Which summary is the better one? An excellent summary is coherent, accurate, concise, and detailed. Answer with A or B. | |
| | Answer: | |

## B.5 Summarization Ranking Procedure Details

We use a standard ranking scheme where each of $K$ summaries is given a rank between 1 and $K$ (inclusive). Sometimes refinements are exact copies of the initial summaries or are very similar in terms of quality, which is why we allow for summaries to be tied. When calclating the win rate we assign 0.5 wins for tied samples. We assign the rank $r'$ to all summaries ranked in a tie, where $r' = \frac{r+(r+n-1)}{2}$, $r$ is the rank of the tied elements, and $n$ is the number of ties at the rank. For example, we map a ranking of $(1, 2, 2, 4, 5) \rightarrow (1, 2.5, 2.5, 4, 5)$ and a ranking of $(1, 2, 3, 3, 3) \rightarrow (1, 2, 4, 4, 4)$.

## B.6 Dataset Collection and Analysis

**Annotation process** To ensure the high quality of our human annotations, we employ experienced annotators sourced through the data-labeling company Surge AI. During an onboarding and evaluation process, we calculate author-annotator agreement on the binary comparison task and manually review the quality of the written feedback and ideal summaries to ensure their high quality. Then we select 31 qualified annotators for all annotation tasks, though they can choose which tasks to participate in and for how long. To further ensure the quality of our annotations, we provide detailed instructions, which we provide to the annotators, and update throughout the process to ensure continuous improvement (these instructions can be found in Appendix C). To measure the agreement rate between the annotators and the authors, we select a sample of 10 Reddit posts from the training dataset as a gold standard and have 17 annotators label them. When comparing the binary comparison annotations with our own ones, this results in an author-annotator agreement rate of 81.0%. We also calculate the average agreement rate between all the possible annotator combinations, yielding an annotator-annotator agreement of 70%. By utilizing these thorough processes and evaluations, we can ensure the accuracy and reliability of our human annotations.

**Dataset Analysis** The feedback we collect typically addresses the most critical shortcomings of the summaries. In 92.0% of our train samples, the annotators' feedback was complete and addressed all important shortcomings of the summary, as reported by the annotators. Across our train dataset, we observe that the majority of the feedback pertains to coverage (77.0%), with smaller percentages relating to accuracy (16.0%), coherence (5.0%), and other categories (2.0%). We also analyze the length of the various summaries and feedback, measured in the average number of tokens. Our human-written summaries have an average length

of $41.0 \pm 0.1$ tokens, the extracted human summaries from Reddit had an average length of $32.5 \pm 0.1$ tokens, the initial summaries generated by FeedME have an average length of $29.3 \pm 0.1$ tokens, and the feedback written by annotators on these initial summaries has an average length of $20.4 \pm 0.2$ tokens.

In addition to these analyses, we also measure the time it takes annotators to complete various tasks (i.e., binary comparison, feedback writing, and ideal summary writing) on our development dataset. We ignore outliers and consider only samples with annotation times of at least 20 seconds and at most 420 seconds (7 minutes). Annotators take $61.5 \pm 5.3$ seconds on average on the binary comparison task, $182.5 \pm 6.3$ seconds on the feedback task, and $195.5 \pm 6.1$ seconds on the ideal summary task. We plot the annotation times on the development dataset for the tasks of annotating binary comparisons, writing feedback, and writing ideal summaries as histograms in Fig. 9. The annotators are much faster at annotating binary comparisons than feedback or ideal summaries. Writing feedback takes less time than writing ideal summaries, which is expected, as critiquing a task is usually easier than solving it. These comprehensive evaluations demonstrate the high quality and thoroughness of our dataset and annotation processes.

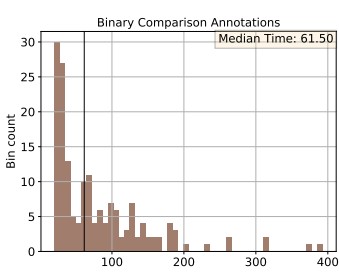
(a) Binary comparison task.

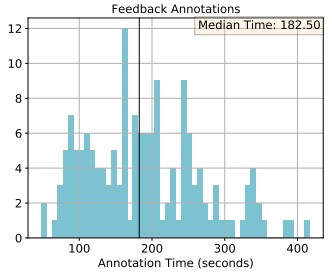
(b) Feedback annotation task.

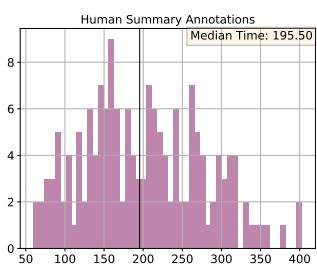
(c) Summary writing task.

Figure 9: Histogram Plot of annotation times (in seconds) of the (9a) binary comparison task, (9b) feedback annotation task, and (9c) the human summary writing task. The evaluation is conducted on the development dataset. We observe that annotators are much quicker at the binary comparison task, which is expected. The results also show that writing feedback takes less time than writing an ideal summary.

## C Annotator Instructions

Overall we completed many annotations to create datasets and evaluate our algorithm. The instructions were task-specific and also continuously updated. In the following, we provide the instructions we used to create our train dataset and the instructions we provided for evaluating the summary quality (of 6 summaries). We will not share more instructions for brevity but can provide them upon request.

### C.1 Train Dataset Annotation Instructions

**Task Overview**

You are given a Reddit Post, which you first need to read carefully. You then need to complete 5 subtasks which consist of comparing two summaries, writing feedback on a summary, classifying the type of feedback, indicating whether there is additional Feedback, and writing an ideal summary. When doing these tasks, please adhere to the guidelines below.

**What makes for a good summary?** Roughly speaking, a good summary is a short piece of text that has the essence of the original text. A good summary tries to accomplish the same purpose and conveys the same information as the original text. We would like you to consider these different dimensions of summaries:

**Essence**: Is the summary a good representation of the post? How well does the summary cover the important information in the post?

**Clarity**: Is the summary reader-friendly? Does it express ideas clearly?

**Accuracy**: Does the summary contain the same information as the post?

**Purpose**: Does the summary serve the same purpose as the original post?

**Concise**: Is the summary short and to the point?

**Style**: Is the summary written in the same style as the original post?

Generally speaking, we give higher weight to the dimensions at the top of the list. The evaluation can be complicated though, since none of the above dimensions are simple yes/no matters, and there aren't hard and fast rules for trading off different dimensions. Use your best judgment and common sense to make these trade-offs. In case the subreddit, title, and Reddit post leave open some ambiguity about what happened, it is important to accurately reflect that in your annotations and not just interpret the text in a certain way. Always look at all the subreddit, title, and Reddit Post and use all information given to make your judgments (sometimes the title may contain crucial information that does not appear in the post but should nevertheless be used).

First, read the Subreddit category, title, and post carefully. A Subreddit is a forum dedicated to a specific topic on the website Reddit. Take your time with this step and re-read the parts that you might not have understood at first. Below is a detailed description of the task you will need to complete for each Reddit post.

Below is a detailed description of each task you will need to complete for each Reddit post:

1. **Comparison Task**: Given a pair of summaries, indicate which is better.

   *Details*: Use the above description of what makes a good summary. It is alright to choose either summary if both summaries are identical copies of each other or if there is no distinguishing feature that makes one summary superior to the other. However, if there is a small detail that makes one summary better than the other, that is enough reason to select that summary.

2. **Feedback Task**: Write short and simple feedback on the given summary about the single, most important shortcoming of the summary. The feedback should NOT mention what category (Accuracy, Coverage, Coherence, other) the feedback belongs to, nor should it assume knowledge about the definitions of "Coverage", "Accuracy", or "Coherence" (see below). Otherwise, the feedback should be as short and simple as possible while still addressing the most important shortcoming of the summary.

   *Details*: You can write the feedback in one or several sentences, but it should only address the single, most important shortcoming of the summary and be as short as possible. There are no other restrictions as to how you write the feedback and what exactly it addresses. If there are no shortcomings in the summary, the feedback can also mention a positive thing about the summary. Use the description of what makes a good summary to trade off the various dimensions that make for a good summary. Often the feedback will (but does not have to) address one of the following axes.

   - **Coverage**: For this axis, answer the question, "how well does the summary cover the important information in the post?" A summary has good coverage if it mentions the main information from the post that's important to understand the situation described in the post. A summary has poor coverage if someone reading only the summary would miss several important pieces of information about the situation in the post. A summary with good coverage should also match the purpose of the original post (e.g., to ask for advice).
   - **Accuracy**: For this axis, answer the question, "does the factual information in the summary accurately match the post?" A summary is accurate if it doesn't say things that aren't in the article, doesn't mix up people, and is generally not misleading. If the summary says anything at all that is not mentioned in the post or contradicts something in the post, it is NOT accurate.
   - **Coherence**: For this axis, answer the question, "how coherent is the summary on its own?" A summary is coherent if, when read by itself, it's easy to understand and free of English errors. A summary is not coherent if it's difficult to understand what the summary is trying to say. Generally, it's more important that the summary is understandable than being free of grammar errors.

Additional Rules: The feedback should NOT mention what category (Accuracy, Coverage, Coherence, other) the feedback belongs to, nor should it assume knowledge about the definitions of "Coverage", "Accuracy", "Coherence", or "other" (as defined above). Example: One should NOT write "This is missing in the area of coverage", or "This summary lacks in the category of accuracy, because ...". The feedback should be understandable to a person who has never read the definition of "Coverage", "Accuracy", and "Coherence". You are, however, ALLOWED to use those words if they make sense on their own, e.g., you CAN say, "This summary does not cover the important parts of the text because", or "This summary is inaccurate as it states ...", or "This is not a coherent summary because ...".

3. **Feedback Type Task**: If your feedback falls into the categories Accuracy-related, Coherence-related, or Coverage-related, mark it as such by checking the corresponding checkbox for the (single) category it is related to. If your feedback is not related to any of these three categories, then check the "Other" checkbox.

4. **More Feedback Task**: Answer with Yes if there is additional Feedback about an important shortcoming of the summary that you would want to mention and No otherwise.

5. **Ideal Summary Task**: Ideal Summary Task: Write a short summary for the Reddit post that is ideal in your view.

   *Details*: The ideal summary should be ideal in terms of all the criteria mentioned above, i.e., essence, clarity, accuracy, coverage, purpose, conciseness, coherence, and style. In other words, you should not be able to find an obvious critique of the ideal summary that you write. It is okay to reuse parts of previous summaries but only if those parts should be a part of an ideal summary. The ideal summary should maximally be 48 tokens long (otherwise, you can't submit your annotation). Tokens are generated by taking your ideal summary and splitting up certain words into individual pieces (this is necessary to train our AI). The interface will show you how many tokens your ideal summary has already taken up.

## C.2 Summary Quality Evaluation Instructions

**Task Overview**

You will be given a Subreddit category, a title, and a Reddit Post, which you first need to read carefully. Your task is then to compare 6 summaries and rank them according to quality.

**What makes for a good summary?** Roughly speaking, a good summary is a short piece of text that has the essence of the original text. A good summary tries to accomplish the same purpose and conveys the same information as the original text. We would like you to consider these different dimensions of summaries:

**Essence**: Is the summary a good representation of the post? How well does the summary cover the important information in the post?

**Clarity**: Is the summary reader-friendly? Does it express ideas clearly?

**Accuracy**: Does the summary contain the same information as the post?

**Purpose**: Does the summary serve the same purpose as the original post?

**Concise**: Is the summary short and to the point?

**Style**: Is the summary written in the same style as the original post?

Generally speaking, we give higher weight to the dimensions at the top of the list. The evaluation can be complicated though, since none of the above dimensions are simple yes/no matters, and there aren't hard and fast rules for trading off different dimensions. Use your best judgment and common sense to make these trade-offs. In case the subreddit, title, and Reddit post leave open some ambiguity about what happened, it is important to accurately reflect that in your annotations and not just interpret the text in a certain way. Always look at all the subreddit, title, and Reddit Post and use all information given to make your

judgments (sometimes the title may contain crucial information that does not appear in the post but should nevertheless be used).

First, read the Subreddit category, title, and post carefully. A Subreddit is a forum dedicated to a specific topic on the website Reddit. Take your time with this step and re-read the parts that you might not have understood at first. Below is a detailed description of the task you will need to complete for each Reddit post.

**Comparison Task**: Given 6 summaries, indicate which is better by ranking them according to quality. Rank 1 is considered the highest rank, and Rank 6 is considered the lowest rank. The summary with the best quality should be ranked highest, i.e., as Rank 1, and the summary with the worst quality should be ranked lowest, i.e. Rank 6. Use the above description of what makes a good summary. Ties between summaries are allowed, but only if summaries are exact copies of each other or if there is no distinguishing feature that makes one summary superior to the other. However, if there is a small detail that makes one summary better than the other, that is enough reason to rank that summary as better than the other summary. We use Standard Competition ranking (i.e., example rankings of 122456). In standard competition ranking, items that compare equally receive the same ranking number, and then a gap is left in the ranking numbers. The number of ranking numbers that are left out in this gap is one less than the number of items that are compared equally. Equivalently, each item's ranking number is 1 plus the number of items ranked above it.

## C.3   Reward Model

Here we describe the various RMs that we evaluate in more detail. We evaluate the final RM that we use, which produces a language output (e.g., " Yes" or " No") and a standard reward model that produces a scalar output.

**Standard RM.**   Akin to Stiennon et al. (2020), we remove the last embedding layer of a language model and train it to output a scalar value. This scalar value predicts which summary, $x \in \{x_0^0, x_0^1\}$, is better as judged by a human, given a context $c$. We use the OPT 13B LM, introduced in (Zhang et al., 2022), as the base model for our RM and fine-tune it on the human preference comparisons that we collected. It is worth noting that it is not possible to add linear layers on top of GPT-3 models provided via the API, which is why we use the OPT model.

**Reward Model with Language Output.**   In addition to the classic RM (Stiennon et al., 2020), we train an RM to output language tokens instead of a scalar value. To do so, we fine-tune an LM to classify whether a summary $x_0$ is high quality or not, by training it to predict a label $y \in \{y_{good}, y_{bad}\}$. We then fine-tune the LM to maximize $\lambda \log p(x_0) + \log p(y|x_0)$, where $\lambda \in [0, 1]$, chosen using the development dataset. The complete loss can also be written as:

$$\mathcal{L}(p_\theta, x, y) = -\lambda \cdot \sum_{t=1}^{|x|} \log p_\theta(x_t|x_{<t}) - \sum_{t=1}^{|y|} \log p_\theta(y_t|x, y_{<t}).$$

where the subscript $t$ indicates the token index. We evaluate the fine-tuned LM on a given summary $x_0$ by computing $p(y_{good}|x_0)$. The best RM overall uses the following instruction *Is the above an excellent summary of the given text? An excellent summary is coherent, accurate, concise, and detailed. Answer with Yes or No.*, which we refer to as the OPT-RM (when fine-tuning OPT-13B) and GPT-3 Binary (when fine-tuning GPT-3-175B). We also explore fine-tuning on another prompt, where we provide both summaries $A$ and $B$ to the LM and instruct it to indicate which summary is preferred, i.e. *Question: Which summary is the better one? An excellent summary is coherent, accurate, concise, and detailed. Answer with A or B.* We then fine-tune the LM on the label of the preferred summary (according to binary human feedback), i.e. on $y \in \{y_A, y_B\}$. We evaluate the fine-tuned LM on a given summary $x_0$ by computing $p(y_A|x_0)$. We refer to this RM as *Comparison* RM. We explore two RMs, namely, OPT-13B Zhang et al. (2022), and GPT-3-175B and refer to Appendix C.4 for the hyperparameters we use and to Appendix B.4 for the prompt templates).

**Results**   We evaluate all RMs on our validation dataset, and calculate the accuracy of predicting the preferred summary out of two, based on human preferences. Table 15 shows the complete results, and here

we report on some of the RMs trained on 5K samples. The OPT model with the standard RM loss achieves an accuracy of $71.8 \pm 2.0\%$ on the validation dataset. The results further show that both of our methods for training OPT with the LM loss outperform the standard RM loss, with OPT comparison achieving an accuracy of $72.6 \pm 1.9\%$, and OPT-RM an accuracy of $73.4 \pm 1.9\%$. We obtain similar results with fine-tuning GPT-3-175B, achieving an accuracy of $71.2 \pm 2.0\%$ with the GPT3 Comparison, and an accuracy of $74.2 \pm 2.0\%$ with GPT-3 Binary, which outperforms the OPT-RM.

Based on these results, we further evaluate the OPT Binary and GPT-3-175B Binary models on the development dataset that we use to evaluate the scoring functions in §4.3. We calculate the fraction of times the refinement selected by an RM is better than a randomly-selected refinement ("win rate"), according to a ranking given by human evaluators (see App. B.5 for more details). The results can be found in Table 14. OPT-RM achieves a win rate of $63.3 \pm 2.7\%$, and the GPT-3-175B Binary model achieved a win rate of $61.8 \pm 2.9\%$. In this evaluation, OPT-RM outperforms GPT-3 Binary. When considering the results from both the validation and development datasets, both OPT-RM and GPT-3-Binary seem to perform similarly. Given that we have more control over the training process of OPT, the possibility of releasing the model, and the cost involved in training using OpenAI's API, we select OPT-RM model as our reward model for comparison with ILF. In Figure 10, we show the validation accuracy of OPT-RM trained on 100, 1K, and 5K samples on a log-log plot. The figure shows scaling when increasing the dataset size.

We further evaluate results for fine-tuning OPT-RM on the dataset of Stiennon et al. (2020), and also evaluating their model with 1.3B parameters on our dataset. We observe that the binary preference distribution of the training dataset has a significant impact on the performance of the reward model. For example, OPT-RM trained on 5K samples of our own train dataset (i.e., our final reward model) achieves an accuracy of $61.9 \pm 0.2\%$ on the test set from Stiennon et al. (2020) (not shown in Table 15). When this same model is trained on 90K samples from the train dataset of Stiennon et al. (2020), it achieves an accuracy of $69.3 \pm 0.2\%$ on their test set (also not shown in Table 15). In contrast, this same model trained on 90K samples from their train dataset achieves an accuracy of only $68.6 \pm 2.0\%$ on our validation dataset, which is significantly lower than the accuracy of $73.4 \pm 1.9\%$ achieved by the model trained on 5K samples of our own train dataset. Similar patterns can be observed when comparing the OPT Binary model with 1.3B parameters trained on 5K samples of our own train dataset to the released 1.3B reward model trained by Stiennon et al. (2020) on approx. 64K samples of their own train dataset. The former model achieves an accuracy of $69.6 \pm 2.0\%$ on our validation dataset, while the latter only achieves an accuracy of $63.8 \pm 2.1\%$ (note, though, that the RMs are trained with different loss functions). These results highlight two important considerations: (1) preference distributions can vary significantly and have a strong effect on what a reward model learns, and (2) the sample efficiency of a reward model depends heavily on the train and test distributions. If the test distribution differs from the train distribution, reward models may be very sample inefficient and fail to accurately learn the true distribution, even when given significantly more samples.

Table 14: In a human evaluation, we compare reward models and ranking methods on the development dataset (in the same way as in Fig 5). Both RMs are trained on 5K samples and outperform the zero-shot methods.

|  | Scoring Function | Win Rate vs Random Selection (in %) |
|---|---|---|
| Task Specific Heuristic | Max Length | $65.0 \pm 2.7$ |
| Zero-Shot | Embedding Similarity | $48.3 \pm 3.0$ |
|  | **InstructRM Ensemble** | **$56.0 \pm 3.0$** |
| Fine-tuning on 5K samples | **OPT Binary** | **$63.3 \pm 2.7$** |
|  | GPT-3 Binary | $61.8 \pm 2.9$ |

Table 15: In a human evaluation, we evaluate various RMs on the development dataset and validation dataset. We also report the results of training on the train dataset of Stiennon et al. (2020) and evaluating on our development and validation datasets. We calculate the accuracy of predicting which of two summaries is preferred by a human.

| | Models | No. Params | Training Dataset Size | Dev. Acc. (%) | Val. Acc. (%) |
|---|---|---|---|---|---|
| LM Loss / Our dataset | OPT Comparison | 13B | 5K | $66.5 \pm 3.3$ | $72.6 \pm 1.9$ |
| | OPT RM | 1.3B | 5K | $70.0 \pm 3.2$ | $69.6 \pm 2.0$ |
| | OPT RM | 13B | 100 | $54.5 \pm 3.5$ | $53.4 \pm 2.2$ |
| | OPT RM | 13B | 1K | $68.5 \pm 3.2$ | $67.2 \pm 2.1$ |
| | **OPT RM** | **13B** | **5K** | $\mathbf{69.5 \pm 3.2}$ | $\mathbf{73.4 \pm 1.9}$ |
| | GPT-3 Comparison | - | 5K | 68.0 | $71.2 \pm 2.0$ |
| | **GPT-3 Binary** | - | **5K** | - | $\mathbf{74.2 \pm 2.0}$ |
| RM Loss / Our dataset | OPT | 13B | 5K | $68.5 \pm 3.2$ | $71.8 \pm 2.0$ |
| RM Loss / Stiennon et al. (2020) train dataset | Stiennon et al. (2020) RM | 1.3B | 64K | $58.0 \pm 3.4$ | $63.8 \pm 2.1$ |
| LM Loss / Stiennon et al. (2020) train dataset | OPT Binary | 13B | 90K | $69.0 \pm 3.2$ | $68.6 \pm 2.0$ |

## C.4 ILF for Text Summarization Hyperparameters

### C.4.1 Generating Refinements

For all summarization experiments we sample up to 48 tokens (as in Stiennon et al., 2020) with nucleus sampling Holtzman et al. (2019) with $p = 0.95$ and temperature $t = 1.0$. We strip non-alphanumeric characters (e.g., newlines) from the beginning of sampled summaries. We further remove empty white spaces in the generated summaries and remove all text that comes after a new line token $/ n$. Due to the maximum token length, sampled summaries sometimes end with incomplete sentences. Thus, we remove ending sentences that do not end in ".", "!", or "?". The described temperature and post-processing are applied to all summary generations, i.e., for generating initial summaries, refinements, and test summaries.

## C.5 Fine-tuning on Summaries

We conduct independent hyperparameter optimization sweeps with three dataset sizes of human summaries of 100, 1K and 5K samples, and then use the same hyperparameters for fine-tuning on refinements (ILF) and fine-tuning on initial summaries. We choose to run the hyperparameter sweep on Human summaries since this will not give an unfair advantage to our algorithm that fine-tunes on refinements. For the sweep, we utilize the train dataset of human summaries (consisting of 100, 1K, and 5K samples) and evaluate on the development dataset. Unfortunately, the OpenAI API only provides validation loss and token accuracy for batches of the development dataset, making it impossible to evaluate the model on the full development dataset during training. As a result, we utilize the model API to evaluate on the full development dataset after fine-tuning and calculate the perplexity of the generated summaries as a performance measure.

Table 16: We report the chosen hyperparameters of fine-tuning on 100, 1K, and 5K samples of HUMAN SUMMARIES. A * symbol indicates that this hyperparameter is optimal but used only for fine-tuning on HUMAN SUMMARIES. For fine-tuning on REFINEMENTS and INITIAL SUMMARIES we inadvertently use the prompt loss weight 0.

| Samples | Epochs | Prompt Loss Weight | Learning Rate |
|---------|--------|--------------------|---------------|
| 100 | 1 | 0 | 0.05 |
| 1K | 1 | 0.05* | 0.02 |
| 5K | 1 | 0.1 | 0.2 |

To determine the optimal hyperparameters, we perform a sweep over a range of values for the following parameters: *epochs* $\{1, 2, 3, 4\}$, *prompt loss weight* $\{0, 0.01, 0.05, 0.1\}$, and *learning rates* $\{0.02, 0.05, 0.1, 0.2\}$. We first sweep over epochs and select the best value, then perform a sweep using that value for the prompt loss weight, and so on. Our empirical observations indicate that the number of epochs has the greatest impact on perplexity, with training for more than one epoch resulting in overfitting. The selected hyperparameters can be found in Table 16.

During the fine-tuning phase for the REFINEMENTS and INITIAL SUMMARIES datasets with 1K samples each, we made an error in our hyperparameter selection. Instead of using a prompt loss weight of 0.05, we mistakenly used a value of 0, when fine-tuning on human summaries. While this error may have slightly impacted our results, the difference in perplexity between the two settings is minimal, with a value of 6.68 for a prompt loss weight of 0.05 and 6.71 for a prompt loss weight of 0. Despite this mistake, our method still outperforms fine-tuning on human summaries for 1K samples, as well as fine-tuning on initial summaries using suboptimal hyperparameters.

## C.6 Multiple Iterations of ILF for Text Summarization

To evaluate multiple iterations of ILF, i.e., multiple iterations of refining-and-fine-tuning, we fine-tune GPT-3-175B on a refinement dataset with 200 and 300 samples. Thus we conduct a hyperparameter optimization on a train dataset of 200 and 300 refinements and evaluate on a development dataset of 200 refinements (instead of human summaries). To determine the optimal hyperparameters, we perform a sweep over a range of values for the following parameters: *epochs* $\{1, 2, 3, 4\}$, *prompt loss weight* $\{0, 0.01, 0.05, 0.1\}$, and *learning rates* $\{0.02, 0.05, 0.1, 0.2\}$. We first sweep over epochs and select the best value, then perform a sweep using that value for the prompt loss weight, and so on. For fine-tuning on 200 refinements we select the following hyperparameters: epochs = 1, prompt loss weight = 0.05, learning rate multiplier = 0.1. For fine-tuning on 300 refinements we select epochs = 1, prompt loss weight = 0, and learning rate multiplier = 0.2. The results are reported in Section C.8.2.

## C.7 Fine-tuning Reward Models

**OPT Reward Model.** For fine-tuning the OPT Reward Model, we perform bayesian hyperparameter optimization for each of the three different types of reward models: *Standard*, *Comparison* and *Classification* (see section C.3). We sweep over the learning rate in the range of $[1e^{-5}, 1e^{-6}]$ and the batch size $\{32, 64\}$ for all the models. For the reward models using the language loss, we also optimize the prompt-loss weight $\{0.0, 0.01, 0.05, 0.1, 0.5, 1.0\}$. We run 10 iterations per model and evaluate all the sweeps with the 200 development examples. We use a linear learning rate scheduler and a weight decay of 0.1 for all the runs. The optimal batch size is 32 for all the models. The best prompt loss weight is 0.01 for both the *Comparison* and *Classification* RMs. As for the learning rate, we use $9.3e^{-6}$ for the *Standard* RM, $5.8e^{-6}$ for the *Classification* RM and $1e^{-6}$ for the *Comparison* RM. In the final fine-tuning, we select the best RM in the validation split over 10 epochs.

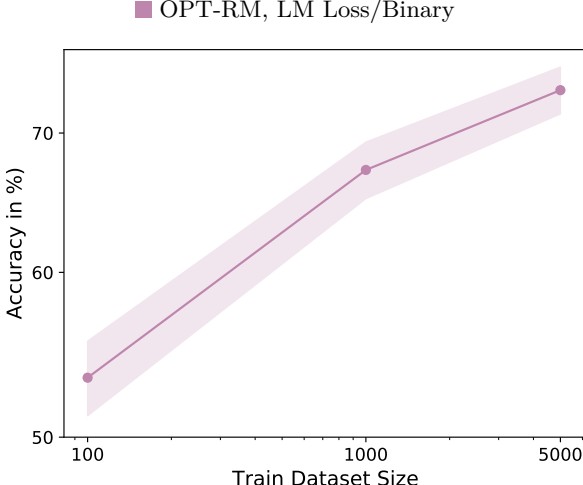

Figure 10: The validation accuracy versus training dataset size of OPT-RM trained on 100, 1K, and 5K samples on a log-log plot.

**GPT-3 Reward Model.** In order to fine-tune GPT-3-175B as an RM, we utilize the OpenAI API. We fine-tune two types of RMs: the *Comparison* RM, which learns to predict which of two summaries is superior, and the *Classification* RM, which predicts whether a given summary is of high quality or not. For cost considerations, we conduct hyperparameter tuning on a training dataset of 1K samples (instead of 5K) and evaluate on a development dataset of 200 samples. We use a dataset with 1K samples for cost reasons. We then apply the same hyperparameters when fine-tuning on 5K samples while implementing early stopping in terms of epochs. Due to the binary nature of the human preference annotations in the classification reward model, the effective train dataset size for this model is doubled to 2K samples.

In order to determine the optimal hyperparameters, we perform a sweep over a range of values for the number of epochs $\{1, 2, 3, 4\}$ and the prompt loss weights $\{0, 0.001, 0.005, 0.01, 0.05, 0.1, 0.5\}$. The OpenAI API provides classification accuracy (for both the comparison and classification tasks) for the full development dataset after each epoch, allowing us to select the appropriate number of epochs and prompt loss weight. When fine-tuning on 5K samples, we utilize early stopping to prevent overfitting, using 1 epoch and a prompt loss weight of 0 for the comparison model and 4 epochs and a prompt loss weight of 0.001 for the classification model. We use default values for all other hyperparameters, which may vary depending on the dataset size.

## C.8 Additional Text Summarization Results

### C.8.1 Results: ILF + OPT-RM

In this section, we present the full results of our best-performing method ILF + OPT-RM and other additional methods (see §4.4.1 for a description of ILF + OPT-RM and §4.4.3 for a discussion of the results). We conduct the same evaluation as described in §4.4.2, i.e. in a human evaluation, annotators rank various test summaries based on quality. We then calculate the win rate against human written summaries, which we use as an evaluation metric. Importantly, all methods evaluated here are trained on datasets with 5K samples. Note that the methods compared here are not exactly the same as the methods compared in Fig. 6. Concretely, the test summaries generated by the methods fine-tuning on refinements (ILF), fine-tuning on human summaries, and OPT-RM best-of-64 FeedME are the same as in Fig. 6, for the test summaries generated by corresponding methods trained on 5K samples. Here, however, we don't evaluate FeedME and fine-tuning on initial summaries. However, we evaluate ILF + OPT-RM (best-of-64), our best-performing model, which we also added to Fig. 6 for reference. We also evaluate a new method called *Fine-tuned on Feedback + Refinements*, which we describe below.

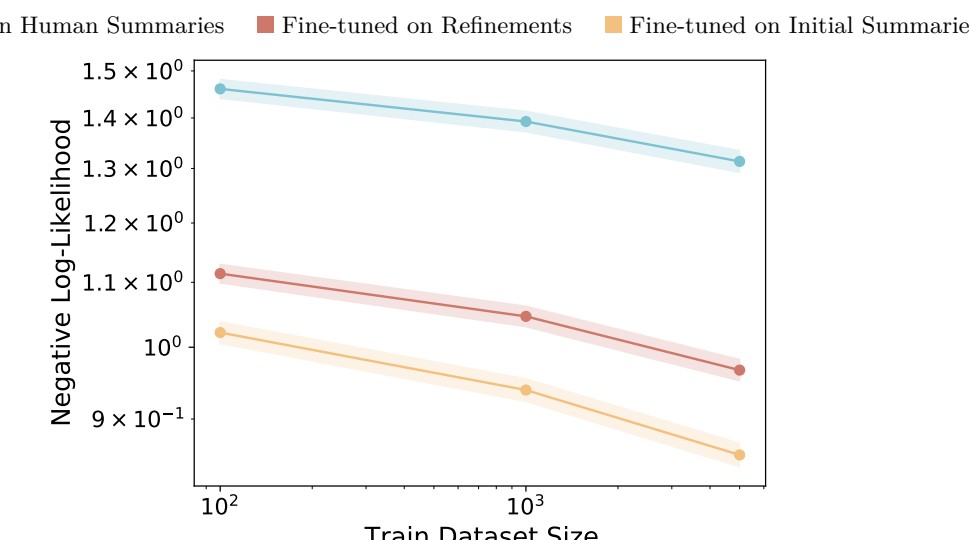

Figure 11: Evaluation of models fine-tuned on 5K initial summaries, refinements, and human summaries on 500 samples from the corresponding validation datasets. For example, the model fine-tuned on human summaries is evaluated on 500 human summaries from the validation dataset. The model fine-tuned on refinements has a significantly lower negative log-likelihood than the model fine-tuned on human summaries.

For fine-tuning on feedback + refinements, we use a title, post, and summary as input and the model is trained to predict the corresponding feedback and refinement. Our motivation for this approach is that generating feedback first may improve the quality of the resulting refinements, similar to the findings of previous work on self-prompting methods Saunders et al. (2022); Bai et al. (2022b) and the Chain of Thought (CoT) prompting technique Wei et al. (2022b). CoT has been shown to improve the performance of models across various tasks Wei et al. (2022b) when allowing the model to reason before answering a question. For fine-tuning on feedback and refinements, we utilize the initial summaries that were used to gather human feedback, as well as the refinements generated by our method. We use the loss $\log p(x_1, f|\text{prompt}) + \lambda \log p(\text{prompt})$, i.e. we learn to predict the refinement and the feedback. We employ the same hyperparameters as in the fine-tuning on refinements algorithm (including the prompt loss weight). During testing, we require initial summaries, from which we generate feedback and refinements. As initial summaries, we use the test samples generated by FeedME (as evaluated in Figure 6). To ensure compatibility with the 48-token length restriction of the test summaries, we append the special end token / n ### to the end of the feedback and refinements during training. At test time, we set the maximum number of tokens to generate 300, and terminate generation when the stop-word / n ### appears. We then apply the same postprocessing procedure outlined in Appendix C.4.1 to shorten the refinements to 48 tokens. We refer to Appendix B.3 for the exact prompt templates we used.

We present all the results in Fig. 12. We find that fine-tuning on a set of 5K refinements achieves a win rate of $36.0 \pm 1.8\%$, while ILF + OPT-RM (best-of-64) has a win rate of $50.8 \pm 1.9\%$, achieving human-level summarization performance (see §4.4.3 for a more detailed discussion). OPT-rM best-of-64 FeedMe achieves a win rate of $45.1 \pm 1.9\%$, fine-tuning on a set of 5K human-generated summaries achieves a win rate of $35.4 \pm 1.8\%$, and fine-tuning on a combination of 5K feedback and refinements has a win rate of $26.1 \pm 1.7\%$. It is worth noting that the performance of fine-tuning on feedback and refinements is lower than that of fine-tuning on refinements alone. We attribute this to the increased difficulty of generating both feedback and refinements and believe that this discrepancy may be due to limitations in our models, dataset size, or hyperparameters. Previous work has demonstrated the feasibility of training models to generate feedback Saunders et al. (2022); Bai et al. (2022b), so we believe that further optimization and experimentation may improve the performance of this method. We further want to note that the results for fine-tuning on 5K refinements, 5K human summaries, and best-of-64 FeedME deviate from the results in Fig 6. This is because

we compare different methods with each other, and human annotations generally contain some amount of noise (given that different people annotate the same samples).

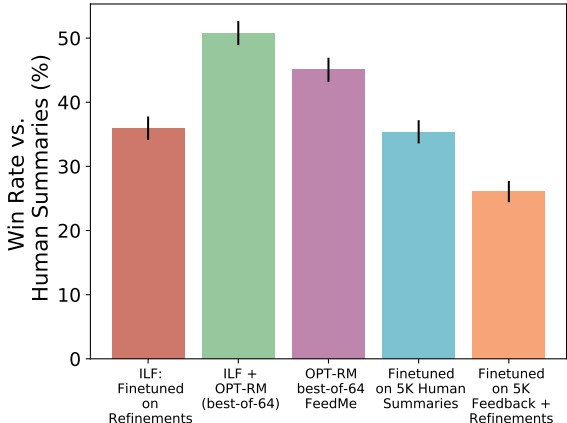

Figure 12: How often human evaluators prefer summaries from ILF: Fine-tuned on Refinements, OPT-RM best-of-64 FeedME, ILF + OPT-RM (best-of-64), fine-tuning on human summaries, and fine-tuning on feedback + refinements (all methods fine-tuned on 5K samples). ILF + OPT-RM (best-of-64) generates summaries of a similar quality to human summaries. Fine-tuning on feedback + refinements performs worse than fine-tuning on refinements (ILF).

### C.8.2 Multiple Iterations of ILF

Our experiments suggest that ILF is an effective method for leveraging language feedback in the training of LMs. Here we explore ILF in its most general form by doing multiple iterations of refining-and-fine-tuning.

**Dataset Improvement.** In this experiment, we evaluate the effectiveness of iterative refinement of the dataset distribution using ILF. To this end, we first fine-tune GPT-3-175B on 100 refinements from iteration 1 of ILF (i.e. doing one iteration of refining initial summaries, as we did in the main results of our paper, see §4.4.2) and refer to this fine-tuned model as $M_1^{100}$. The notation we use here is that the subscript indicates the iteration of ILF that the refinements were generated in, and the superscript indicates the number of overall samples the model is fine-tuned on. We also refer to the dataset of 100 refinements from iteration 1 as $\mathcal{D}_1^{100}$. As a baseline, we fine-tune $M_1^{100}$ on an additional 100 refinements from ILF iteration 1, resulting in $M_1^{200}$, i.e., a model trained on 200 refinements from ILF iteration 1. We then compare this baseline to two iterations of ILF. Specifically, we use $M_1^{100}$ to generate summaries for an additional 100 samples (the same Reddit posts as for the baseline) and collect human feedback on those summaries. We then use this feedback to generate 5 refinements using the FeedME[10] and then select the best refinement using our InstructRM method. We refer to these 100 selected refinements from the second iteration of ILF as $\mathcal{D}_2^{100}$. Finally, we fine-tune $M_1^{100}$ on $\mathcal{D}_2^{100}$ to obtain the model $M_{1,2}^{200}$, which has been trained on a total of 200 refinements generated in both the first and second iterations of ILF. All fine-tuning was performed using the same hyperparameters as described in Appendix C.4 for fine-tuning on 100 refinements. We refer to Table 17 for an overview of all models and train datasets.

In this human evaluation, we compare the performance of the summaries generated by the baseline model $(M_1^{200})$ with those generated by two iterations of ILF $(M_{1,2}^{200})$ on our test set. Human evaluators are asked to indicate their preferred summary for each comparison, and the win rate of $M_{1,2}^{200}$ against $M_1^{200}$ is calculated and plotted in Fig. 13 (left)[11]. Our results show that two iterations of ILF outperform one iteration with a win rate of $53.2 \pm 1.9\%$ indicating that applying multiple rounds of ILF can improve the data distribution. However, we also want to investigate whether multiple rounds of ILF lead to better models than directly

---

[10]Ideally, one would use the same model $M_1^{100}$ to generate the refinements. However, in our case, this is not possible since we fine-tuned GPT-3-175B, which is not an instruction-fine-tuned model.

[11]Note, we set the win rate manually to 50% at 100 samples, since the baseline is equivalent to one iteration of ILF.

Table 17: Datasets (refinements) over which the models $M$ are trained, and which they generate. The superscript indicates the number of samples, whereas the subscript indicates the ILF step. In this figure we do not show FeedME which is used to generate the refinements given feedback.
* these samples are new samples from the interval [100,200] of $\mathcal{D}_1^{200}$

| Initial Model | Fine-tuned Model | Fine-tuning Dataset | | | Produces Dataset |
|---|---|---|---|---|---|
| | | ILF Iter. 1 | ILF Iter. 2 | ILF Iter. 3 | |
| GPT-3-175B | $M_1^{100}$ | $\mathcal{D}_1^{100}$ | – | – | $\mathcal{D}_2^{100}$ |
| $M_1^{100}$ | $M_1^{200}$ | $\mathcal{D}_1^{100*}$, | – | – | – |
| GPT-3-175B | $M_{scratch,1}^{200}$ | $\mathcal{D}_1^{200}$ | – | – | – |
| GPT-3-175B | $M_{scratch,1}^{300}$ | $\mathcal{D}_1^{300}$ | – | – | – |
| $M_1^{100}$ | $M_{1,2}^{200}$ | $\mathcal{D}_1^{100}$ | $\mathcal{D}_2^{100}$ | – | $\mathcal{D}_3^{100}$ |
| $M_{1,2}^{200}$ | $M_{1,2,3}^{300}$ | $\mathcal{D}_1^{100}$ | $\mathcal{D}_2^{100}$ | $\mathcal{D}_3^{100}$ | – |
| GPT-3-175B | $M_{scratch,1,2}^{200}$ | $\mathcal{D}_1^{100} + \mathcal{D}_2^{100}$ | | – | – |
| GPT-3-175B | $M_{scratch,1,2,3}^{300}$ | $\mathcal{D}_1^{100} + \mathcal{D}_2^{100} + \mathcal{D}_3^{100}$ | | | – |

fine-tuning on the same number of refinements from the first round from scratch. In other words, while our current baseline consists of further fine-tuning $M_1^{100}$ on an additional 100 samples, it is also possible to directly fine-tune GPT-3-175B on 200 refinements from the first iteration of ILF from scratch, i.e. $M_{scratch,1}^{200}$. We aim to determine the relative effectiveness of these two approaches in improving model performance on the text summarization task.

**Model Improvement** In this experiment, we aim to compare the performance of multiple rounds of ILF to directly fine-tuning on a comparable number of refinements from the first iteration of ILF. As a baseline, we fine-tune GPT-3-175B on 200 and 300 refinements from the first iteration of ILF and conduct hyperparameter tuning as described in the Appendix C.4. We then compare these baselines to two and three rounds of ILF. For the two-round ILF model, we use the previously described $M_{1,2}^{200}$. To obtain the three-round ILF model, we use $M_{1,2}^{200}$ to generate summaries for an additional 100 samples (on the same Reddit posts as for the baseline), gather human feedback, generate 5 refinements with GPT-3-175B using the feedback, and select the best refinement using InstructRM, resulting in $\mathcal{D}_3^{100}$. We then fine-tune $M_{1,2}^{200}$ on $\mathcal{D}_3^{100}$ to obtain the model $M_{1,2,3}^{300}$. It is important to note that while our baselines fine-tune GPT-3-175B from scratch on 200 and 300 refinements, the models $M_{1,2}^{200}$ and $M_{1,2,3}^{300}$ are obtained by continuously fine-tuning a model iteratively on additional refinements. This difference in approach may introduce a discrepancy in the results, as we use different hyperparameters, and the dataset size may affect the learning dynamics. To control for this potential difference, we also fine-tune GPT-3-175B from scratch on the refinements generated through various iterations of ILF. Specifically, as an alternative to $M_{1,2}^{200}$, we fine-tune GPT-3-175B from scratch on a concatenation of 100 refinements from the first round of ILF (i.e., $\mathcal{D}_1^{100}$) and 100 refinements from the second round of ILF (i.e., $\mathcal{D}_2^{100}$), and refer to the resulting model as $M_{scratch1,2}^{200}$. Similarly, as an alternative to $M_{1,2,3}^{300}$, we fine-tune GPT-3-175B from scratch on a concatenation of 100 refinements from the first round of ILF ($\mathcal{D}_1^{100}$), 100 refinements from the second round of ILF $\mathcal{D}_2^{100}$, and refinements from the third round of ILF (i.e. $\mathcal{D}_3^{100}$), and refer to the resulting model as $M_{scratch1,2,3}^{300}$. It is worth noting that the refinements from the second and third rounds of ILF (i.e. $\mathcal{D}_2^{100}$ and $\mathcal{D}_3^{100}$) are based on summaries generated using models that were continuously fine-tuned (i.e. $M_1^{100}$ and $M_{1,2}^{200}$). As such, the models $M_{scratch1,2}^{200}$ and $M_{scratch1,2,3}^{300}$ are not a direct application of ILF, but rather an approximation of the distribution induced by ILF. We refer to Table 17 for an overview of all models and train datasets.

Using a human evaluation, we compare the performance of the three methods on the test dataset: the baseline, ILF with continuous fine-tuning, and ILF approximated by fine-tuning from scratch. The results

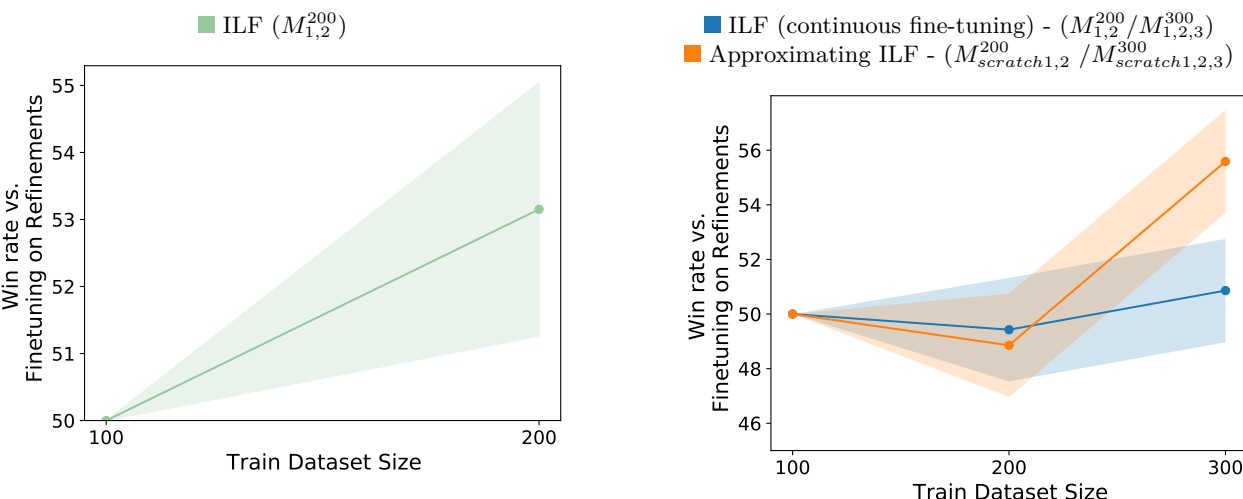

Figure 13: **Left**: Win rate of 2 iterations of ILF against fine-tuning on the same number of refinements from the first iteration of ILF. **Right**: Win rate of 3 iterations of ILF, and approximating 3 iterations of ILF by fine-tuning from scratch, against fine-tuning on the same number of refinements from the first iteration of ILF.

are shown in Fig. 13 (right). With this more realistic baseline, we find that directly applying ILF does not improve upon the baselines, with win rates of $49.4 \pm 1.9\%$ and $50.9 \pm 1.9\%$ for 200 and 300 samples, respectively. However, approximating ILF by fine-tuning from scratch on the distributions induced by ILF significantly improves upon the baseline for 300 samples, with a win rate of $55.6 \pm 1.9\%$. The method is slightly worse than the baseline for 200 samples, with a win rate of $48.9 \pm 1.9\%$. We currently hypothesize that continuous fine-tuning may lead to catastrophic forgetting, while fine-tuning from scratch may not have this problem. This could explain why $M^{300}_{scratch1,2,3}$ performs significantly better than $M^{300}_{1,2,3}$ for 300 samples. Specifically, $M^{200}_{1,2}$ may actually generate an improved distribution in the third iteration of ILF. However, when further fine-tuning $M^{200}_{1,2}$ on this improved distribution $\mathcal{D}^{100}_2$, the model may forget what it learned previously. On the other hand, the model $M^{300}_{scratch1,2,3}$ that learns from scratch on the concatenation of all datasets produced by ILF may actually benefit from the improved dataset distribution because it does not unlearn anything. It is, however, unclear why $M^{200}_{scratch1,2}$ does not benefit from the improved data distribution $\mathcal{D}^{100}_2$. It is also possible that the hyperparameters play a significant role in the final performance of the various models and that the dataset size has a strong influence on model performance (e.g., fine-tuning on more samples may be more stable than fine-tuning on fewer samples). In future work, we plan to conduct more elaborate experiments to answer these questions and better understand the effects of the dataset size and number of iterations on ILF. Specifically, we aim to run multiple iterations of ILF and use $M^{200}_{scratch1,2}$ as the model to generate summaries in the third round of ILF (instead of $M^{200}_{1,2}$). This would be a direct implementation of ILF, rather than an approximation of it, as we would be fine-tuning the same model with which we are also generating an improved distribution. We also hope to investigate the effect of the dataset size and number of iterations on ILF. Overall, our results suggest that ILF has the potential to improve the performance of natural language processing systems by continuously incorporating human feedback into the training of language models, but further research is needed to fully understand the best ways to leverage this approach.

### C.8.3 Part-of-Speech Distribution for Fine-tuning Datasets

We evaluate the negative log-likelihood of GPT-3-175B on the three fine-tuning datasets, i.e. on initial summaries, refinements, and human summaries. We use the training dataset with 1K samples and calculate the negative log-likelihood over different Part-of-Speech tags. We use Stanza Qi et al. (2020) as the PoS tagger for this experiment and then we separate the words into three groups: function words, content words, and others. The function words are words that have little lexical meaning: articles, pronouns, adpositions,

conjunctions, auxiliary verbs, particles and interjections. On the other hand, content words are words that contain semantic information: nouns, adjectives, adverbs and lexical verbs. We keep numbers and symbols under the group *others*. With this analysis, we want to spot different patterns between model-generated (initial summaries and refinements) and human-written summaries. Note that a high negative log-likelihood implies a high loss. We present the results in Fig 14. Since the average loss is higher for human summaries, we normalize all the loss values by transforming them to have mean 0 and standard deviation 1. Overall, the word distribution is very similar for all three fine-tuning datasets. In terms of normalized mean loss, it is interesting how the content words have a bigger influence on the refinements dataset. We believe that this is related to our results in section 4.4.3, where we obtain the best results when fine-tuning on refinements.

### C.8.4 Comparison to Results of Scheurer et al. (2022)

Here we relate our results to our previous non-archival work, Scheurer et al. (2022). In Fig. 2 of Scheurer et al. (2022), they compare their method of fine-tuning on refinements against various baselines, such as fine-tuning on initial summaries, sampling from FeedME (called InstructGPT), and sampling from GPT-3-175B. They calculate the win rate of all methods against human written summaries (Völske et al., 2017) that are automatically extracted from Reddit. As shown in §4.2 and App.B.6, our human summaries are preferred $72.3 \pm 3.2\%$ to the human summaries of Völske et al. (2017). This implies that the win rates in Scheurer et al. (2022) are much higher than in our case since we use a much stronger baseline.

We now present three differences between the results found in Scheurer et al. (2022) and the results found in our paper. Then we will provide various potential reasons that could explain the differences. First, when comparing the results (in relative terms) in Scheurer et al. (2022) Fig. 2 to our results in Fig. 6 where we fine-tune on 100 samples, we see differences in performance. Scheurer et al. (2022) reports that fine-tuning on refinements outperforms fine-tuning on initial summaries. And both methods outperform sampling from FeedME (i.e., InstructGPT). In our experiments fine-tuning on 100 refinements achieves a win rate of $19.6 \pm 1.5\%$ against human summaries, fine-tuning on initial summaries a win rate of $19.6 \pm 1.5\%$, and FeedME a win rate of $20.8 \pm 1.5\%$. Thus both fine-tuned methods perform equally and are worse than sampling from FeedME.

Second, we compare the results of refining a summary with feedback. Note that Scheurer et al. (2022) uses an embedding-based scoring function to select refinements, whereas we use InstructRM. In Scheurer et al. (2022) Fig. 3 (left) REFINE WITH FEEDBACK + BEST OF N achieves a win rate of $67.0 \pm 3.1\%$ against initial summaries (sampled from FeedME), REFINE WITH FEEDBACK achieves a win rate of $60.5 \pm 3.0\%$, REFINE WITHOUT FEEDBACK achieves $50.3 \pm 2.6\%$ and Human Summaries have a win rate of $60.8 \pm 3.4$. In our Fig. 15 (left) Refine with Feedback + Best-of-5 achieves a win rate of $69.1 \pm 1.9\%$, Refine with Feedback achieves a win rate of $63.9 \pm 2.0\%$, Refinement without Feedback achieves a win rate of $59.4 \pm 2.0\%$ and Human Summaries a win rate of $83.2 \pm 1.7\%$. The difference in the human summaries is expected, given that we use better human summaries. The Refinement without Feedback method achieves higher results in our work than in Scheurer et al. (2022).

Third, it is also noteworthy that using the embedding similarity as a scoring function worked well in Scheurer et al. (2022), while it does not work in our setting (see Table 5 and §4.3.1 for a discussion of the results). We believe this is because the feedback we collect is written by many annotators and is thus much more diverse, while in Scheurer et al. (2022), the authors themselves wrote the feedback.

Here we now list various differences in the setup of Scheurer et al. (2022) and our paper, which could all account for the different results.

1. Scheurer et al. (2022) use an embedding similarity as a scoring function, while we use InstructRM Ensemble. Looking at Tab. 5 and the corresponding discussion in §4.3.1, already shows that the methods are very different.

2. The human-written summaries are of much higher quality in our paper than in Scheurer et al. (2022) (see §4.2 and App. B.6)

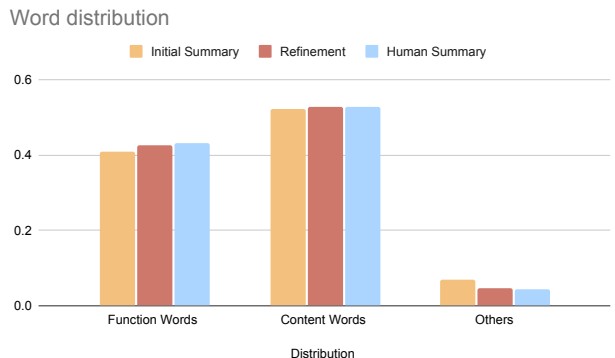
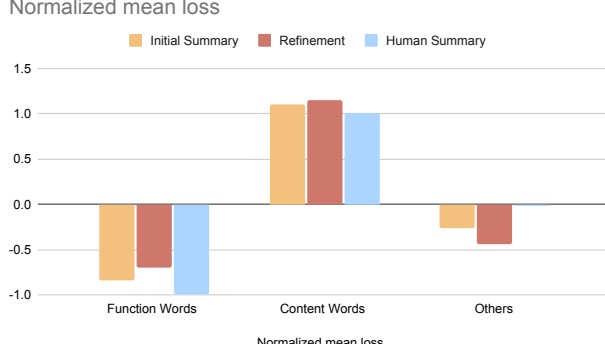

Figure 14: Distribution of tokens of various fine-tuning datasets with 1K samples in terms of content and function words. We only evaluate the various completions, i.e., summaries, since the prompts are the same for all distributions.

3. In Scheurer et al. (2022), the annotation instructions specifically state that the feedback should mention how to improve a summary. In our work, we collect much more unrestricted and diverse feedback. This difference is also apparent in the fact that the embedding similarity does not work well as a scoring function in our setting.

4. In Scheurer et al. (2022), the authors themselves annotated the data, i.e., they wrote the feedback and evaluated the final summaries. In our case, we use independent evaluators who are trained on this task. Using 31 annotators overall also gives us a more diverse and less biased estimate of our methods. Also, doing human evaluations is inherently noisy and will never lead to the exact same results.

5. The evaluation in Scheurer et al. (2022) was done on a different dataset than in this work. Specifically, they used only 100 samples to evaluate their method, while we use a test set of 698 samples.

6. The hyperparameters in Scheurer et al. (2022) used for sampling and fine-tuning are different from the hyperparameters used in our work.

7. Overall, we use different prompts than Scheurer et al. (2022) (see App. B.3 and App. B.1)

### C.8.5 Does Language Feedback Improve Refinements of Summarizations?

The improvements from ILF suggest that the refinements used for fine-tuning are high-quality, so here we investigate whether language feedback is responsible for the high quality. To do so, we have human evaluators rank Refinement with Feedback + Best of N summaries against summaries from several other methods, similar to §4.3. We use the human ranking to compute a win rate between each method and the initial summary. We compare against Refinement with Feedback, which *randomly* chooses a refinement $\in x_1^1, \ldots, x_1^5$. This ablation helps to evaluate the importance of choosing a refinement with our scoring function $R$, i.e., InstructRM Ensemble. We also evaluate Refinement without Feedback, which instructs the LM to refine the initial summary but without feedback. This ablation helps to evaluate the importance of using language feedback. Lastly, we evaluate Human Summaries and Initial Summaries i.e., the initial summary $x_0$ generated by FeedME. We evaluate all methods on the validation dataset.

**Results** Fig. 15 (left) shows the win rates of summaries from various methods against initial summaries. Surprisingly, instructing a model to improve its output without feedback already leads to a significant improvement (win rate of $59.4 \pm 2.1\%$ over the initial summaries). Refinements with Feedback achieve an improved win rate of $63.9 \pm 2.0\%$, showing that language feedback is useful for improving refinement quality. Refinement with Feedback + Best of N achieves an even better win rate of $69.1 \pm 1.9\%$, highlighting that

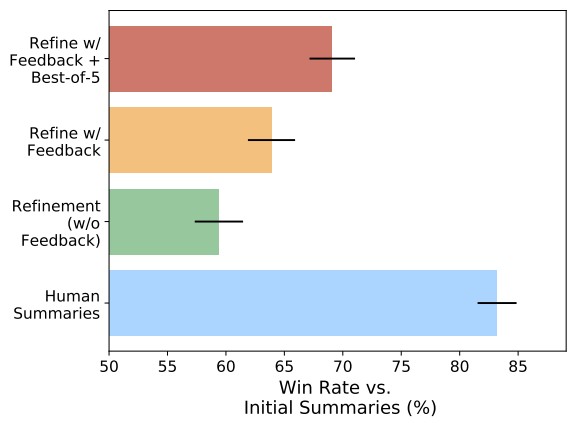 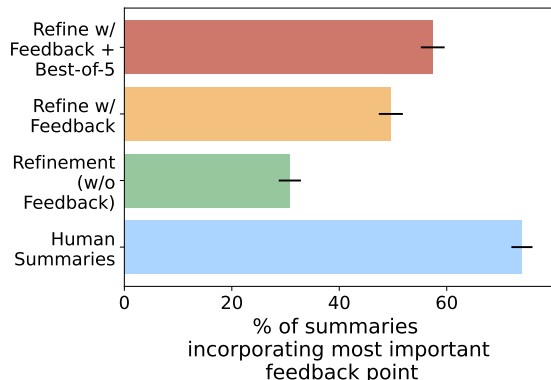

Figure 15: **Left**: Human evaluators prefer summaries from all refinement methods to the initial summaries (FeedME). Refine with Feedback + best-of-5 is rated highest. **Right**: Refine with Feedback + best-of-5 generally does incorporate the most important feedback point.

Best-of-N with the InstructRM Ensemble further improves the refinements. Overall, language feedback is important for high-quality refinements, especially when using Best-of-N sampling.

### C.9 Do Refinements of Summarizations Incorporate the Feedback?

To determine whether refinements are of higher quality due to incorporating feedback rather than improving the summary in other ways, we conduct a study on the validation dataset in which crowd workers evaluate how often the most important point of the feedback is incorporated in the refinements produced by various methods. As shown in Fig. 15, right, our method Refinement with Feedback + Best of N incorporates the most important point in the feedback most frequently ($57.4 \pm 2.2\%$ often). Refinement with Feedback incorporates feedback $49.6 \pm 2.2\%$ of the time, showing that Best-of-N sampling improves how often the feedback is incorporated. For reference, Refinement without Feedback fixes the most important point in the feedback $30.8 \pm 2.1\%$ of the time, despite the model not receiving the language feedback. Human Summaries address the most important point in the feedback $74.0 \pm 1.9\%$ of the time when writing the summary from scratch despite not receiving the feedback explicitly. Our results suggest that refinements are high-quality in part because they incorporate the most important point in the feedback.

# D   Broader Impacts

Our work, like many other human feedback-based machine learning works, relies upon hand-labelled human annotations. Since these annotations are often provided by small groups of expert annotators, the resulting model may exhibit social (Liang et al., 2023; Srivastava et al., 2023), political (Hartmann et al., 2023; Perez et al., 2023), or moral biases (Simmons, 2023) learned from the human annotations. Past work has also shown that such human feedback-based models may have difficulty modeling the full spectrum and diversity of human opinions (Santurkar et al., 2023), and may even rely upon modeling the most productive annotators rather than modeling the task itself (Geva et al., 2019). Continued training on multiple rounds of feedback may also result in further amplification of biases present in the dataset (Taori & Hashimoto, 2023). As such, the deployment of feedback-based models warrants careful attention to the diversity, biases, and steerability of the model.

