# OpenReview forum: "Learning from Natural Language Feedback"
_TMLR — Accepted by TMLR_

### Review · Reviewer_zchu · 2023-11-07

**Summary Of Contributions:**

The paper presents a method for learning from human feedback which uses human-written natural language feedback to improve models, and applies it to code generation (Mostly Basic Python Problems) and summarization (Reddit TL;DR).  The method starts with both natural language feedback and natural language refinements based on that feedback, trains a refinement generator based on that data, then trains a policy on the refinements.

**Audience:**

Yes

**Broader Impact Concerns:**

I don't think a broader impact statement is necessary.

**Claims And Evidence:**

No

**Requested Changes:**

The major changes are for (1) and (3) above:

## Fair presentation of results

The results should be presented more fairly, using correct baselines, easier to understand numbers, and a few table fixes.

1. **Correct baselines:** The correct baseline for ILF on code seems to be "gold standards, human refinements", since this is the simple thing to do with the demonstrations gathered and used by ILF.  Thus, the abstract comparison should to this method, rather than using zero-shot as the baseline.
2. **Easy to understand numbers:** The abstract claims "38% relative (and 10% absolute)" improvement on MBPP.  Ignoring that this is to zero shot, it is better presented as an improvement "from 26% to 36% pass@1 rate".  "relative" and "absolute" are confusing without knowing the raw numbers, even if it possible to back out the raw numbers.
3. **Table fixes:** Several tables are misleading.  Table 3 is particularly unfortunate, since (1) it shows 0% as the baseline without noting that the $\textrm{MBPP}_{\textrm{Train}}$ dataset was explicitly constructed to have 0% by removing examples where the baseline passed and (2) is quoted in the text as being "on the evaluation dataset".  My guess is that the text is simply incorrect, as it would be surprising if the baseline got 0% on the evaluation dataset.  But (1) is also bad: captions should be designed to not mislead a reader who skims the tables, and this one definitely would.  Table 5 is another example: the caption explicitly says "InstructRM Ensemble performs best" and that row is bolded, but two other rows are better: the naive MaxLength baseline is way better, and Prompt 2 is slightly better.  It might still be reasonable to use the ensemble, but it's wrong to say it performs the best.  This is a nonexhaustive list of table problems, so a good through all of them to make them clearer is important.

## Mathematical details

The big one is the importance sampling.  It's bad to say the method is importance sampling when it isn't, because it doesn't divide by the proposal density.  Instead, you should just say that the method approximates $\pi^*(\cdot | t)$ with $q_t$, then uses $q_t$ as a policy improvement operator.  As is, the method is presented as mathematically motivated in a way that it isn't, and the gap between the mathematical presentation and the algorithm is not discussed.

There are a variety of smaller notational issues:

1. The paper says "forming dataset $\mathcal{D} = \\{t\\}$".  This is bad notation, as it looks like a singleton set.
2. It's bad notation to sample $t \sim p(t)$: it should be $t \sim p$ as $t$ isn't known before sampling itself.
3. The $\operatorname{FINETUNE}(\pi_\theta, \mathcal{D})$ is confusing.  The notation for $\mathcal{D}$ makes it seem like it contains task inputs but not outputs, since section 2.2 clearly thinks of $t$ is being the input.  But you can't fine-tune until you have the outputs.
4. Eqns. (6) and (7) are harder to read than just writing out the multistep sampling algorithm, and somewhat misleading since only the sampling algorithm is used, not the density (because the density is intractable).  It would be better to just write the few-line algorithm showing each sampling step.
5. The delta functions used are Kronecker delta functions, not Dirac delta functions.

## Small notes

1. Figure 1 was unclear when I first saw it, as I wasn't sure what the type signature of each component was.  E.g., I was initially expecting there to be a model that generated feedback, but it was missing, and I wasn't sure what the type signature of $\pi_{\textrm{Refine}}$ was.
2. The method shows the model one of the unit tests.  Does it use the rest of the unit tests for rejection sampling?  It's likely I just missed this, but it seems important to state explicitly so that other compares compare to this one accurately.

**Strengths And Weaknesses:**

The method itself is interesting: natural language feedback is very high bandwidth compared to binary preferences or Likert scales.  I also appreciate that human evaluation is used for judging summaries.

However, there are three significant weaknesses:

1. The results are only slightly better than the baselines, but are presented to seem like major baseline improvements.  For example, the abstract says "For code generation, ILF improves a Codegen-Mono 6.1B model’s pass@1 rate by 38% relative (and 10% absolute) on the Mostly Basic Python Problems (MBPP) benchmark, outperforming both fine-tuning on MBPP and fine-tuning on repaired programs written by humans."  The 10% number refers to an absolute improvement from 26% to 36%, but this is comparing their tuning method against against a zero-shot baseline.  It is true that the method also improves against fine-tuning on repaired programs written by humans, but fine-tuning on repaired programs (which is also data used by their method) gets 33% (and is tied at 68% with their method for pass@10).  Reading either the abstract or skimming Table 4 might miss the size of the improvement: the abstract doesn't mention it, and Table 4 explicitly calls "zero-shot" the baseline rather than normal fine-tuning.

2. The results don't demonstrate a human feedback bandwidth improvement.  Indeed, the baselines they compare against use strictly *less* human data, since fine-tuning on refinements alone doesn't need the feedback data.  E.g., section 3.1 notes that "ILF assumes the availability of feedback but not necessarily of the repaired code/refinements, for a variety of reasons", but then notes that they train $\pi_{\textrm{Refine}}$ using human-written refinements.  Thus, the key code result is that using ILF over fine-tuning on demonstrations improves from 33%/68% (pass@1/pass@10) to 36%/68% using strictly more data.

3. The mathematical discussion in section 2 is interesting, but is not very related to the method.  In particular, section 2.2 says they use equation (5) as an approximation, but the method does not actually use equation (5).  Importance sampling as shown in equation (5) requires *dividing* by the proposal distribution density $q_t(x_1)$, but the construction of $q_t$ is sampling only with intractable densities since it's a multi-stage sampling process.  Thus, it would be intractable to divide by $q_t(x_1)$ even if that were desired.  As far as I can tell this issue is not mentioned again in the paper, so the division is silently dropped.

---

> ### Author Response · Authors · 2024-01-05
> **Thank you for your review -- answers to your questions are below, and revisions in the paper are highlighted in red**
>
> We thank the reviewer for their careful and thorough review of our paper -- we have provided responses to their questions/concerns below and revisions/new experiments in the updated manuscript.
> - "The 10% number refers to an absolute improvement from 26% to 36%, but this is comparing their tuning method against against a zero-shot baseline. It is true that the method also improves against fine-tuning on repaired programs written by humans, but fine-tuning on repaired programs (which is also data used by their method) gets 33% (and is tied at 68% with their method for pass@10). Reading either the abstract or skimming Table 4 might miss the size of the improvement: the abstract doesn't mention it, and Table 4 explicitly calls "zero-shot" the baseline rather than normal fine-tuning."
>     - ILF is intended as a method for improving an LLM's outputs without requiring fine-tuning on more gold programs or human-written refinements. As such, our goal is to improve over the zero-shot baseline. It is an added bonus that it improves pass@1 metrics over the fine-tuning methods, particularly the method that trains on human-written refinements. In practice, obtaining human-written refinements for the model outputs is an **impractical and expensive** way to train a model, so we view this method as a gold standard rather than a baseline. We have also added some text to clarify this in Section 3.3.
> - "The results don't demonstrate a human feedback bandwidth improvement. Indeed, the baselines they compare against use strictly less human data, since fine-tuning on refinements alone doesn't need the feedback data. E.g., section 3.1 notes that "ILF assumes the availability of feedback but not necessarily of the repaired code/refinements, for a variety of reasons", but then notes that they train $\pi_\text{Refine}$ using human-written refinements. Thus, the key code result is that using ILF over fine-tuning on demonstrations improves from 33%/68% (pass@1/pass@10) to 36%/68% using strictly more data."
>     - ILF can use any sufficiently powerful $\pi_\text{Refine}$ model, including more powerful models such as `gpt4` that are capable of refining programs based on natural language feedback via few-shot prompting alone, **without further training on human-written refinements** ([Nijkamp et al.](https://arxiv.org/abs/2203.13474), [Joshi et al.](https://doi.org/10.1609/aaai.v37i4.25642)). In fact, we have added experiments in Appendix A.4 where we instead prompt `gpt-3.5-turbo` and `gpt-4` to act as $\pi_\text{Refine}$. In both cases, ILF is still highly effective, significantly exceeding the pass rates of both the baseline and ablations. This demonstrates that **more training data for training $\pi_\text{Refine}$ is not strictly necessary**. Also, as mentioned previously, the key comparison should be against the baseline (improving from 26%/59% to 33%/68% (pass@1/pass@10)), not the gold standard.
>     - Although we originally chose to train our own $\pi_\text{Refine}$ model, we only made this choice due to our compute and API constraints at the time (*i.e.* larger models were difficult to run with our limited compute resources and strict rate limits were imposed by the Codex API). We have revised Section 3.2 to clarify this point.
> - "The mathematical discussion in section 2 is interesting, but is not very related to the method. In particular, section 2.2 says they use equation (5) as an approximation, but the method does not actually use equation (5). Importance sampling as shown in equation (5) requires dividing by the proposal distribution density $q_t(x_1)$, but the construction of $q_t$ is sampling only with intractable densities since it's a multi-stage sampling process..."
>     - Thank you for pointing out this omission -- we have revised Section 2.3 to further explain how we can approximate both $\mathcal{L}_\theta(t)$ and $\mathcal{L}(\theta)$ without direct access to the value of $q_t(x_1)$. (This was in an earlier version of the paper, but was mistakenly deleted in later revisions -- apologies for the oversight!)
>     - In short, we can re-write Eq. 5 as $L_\theta(t)=\underset{{x_1^i\sim q_t}}{\mathbb{E}} \frac{\pi_t^*(x_1^i)}{q_t(x_1^i)}\log\pi_\theta(x_1^i|t)$, where the importance weight $w_i$ is $\frac{\pi_t^*(x_1^i)}{q_t(x_1^i)}$. Due to the consistently high quality of human feedback, we assume that all $x_1\sim q_t$ are of equally good quality. This allows us to then compute $w_i$ by simply computing the un-normalized value of $\pi_t^*$ (*i.e.* $\beta R(x_1^i,t)$ from Equation 2) and using self-normalization instead. For low temperatures, $\beta\to \infty$, which implies that $L_\theta(t)$ is dominated by the highest-reward refinement $x_1^*$ for each task $t$. Then the total loss simplifies to $L(\theta)=-\underset{t\sim p_T}{\mathbb{E}}[L_\theta(t)]\approx -\underset{t\sim p_T}{\mathbb{E}}[\log \pi_\theta(x_1^*|t)]$. This is equivalent to supervised learning over the best refinements per task.

---

> > ### Author Response · Authors · 2024-01-05
> > **Responses continued**
> >
> > - "The abstract claims "38% relative (and 10% absolute)" improvement on MBPP. Ignoring that this is to zero shot, it is better presented as an improvement "from 26% to 36% pass@1 rate"."
> >     - We agree that stating the metric is important -- our abstract says "ILF improves a Codegen-Mono 6.1B model's **pass@1 rate** by 38\% relative (and 10\% absolute)," which we hope is sufficiently clear.
> > - "Table 3 is particularly unfortunate, since (1) it shows 0% as the baseline without noting that the  dataset was explicitly constructed to have 0% by removing examples where the baseline passed and (2) is quoted in the text as being "on the evaluation dataset""
> >     - Regarding (1), we stated in the "Dataset" portion of Section 3.2 that MBPP_Train (as with MBPP_Refine), was designed to include only tasks for which CodeGen did not originally generate any correct programs. To further clarify, we have also added a sentence to Table 3's caption repeating this detail.
> >     - As for (2), in the "Dataset" section we explicitly mentioned that MBPP_Train is used for two purposes -- firstly, to evaluate $\pi_\text{Refine}$'s ability to incorporate feedback into refinements; and secondly, to train $\pi_\theta$ using the correct refinements generated by $\pi_\text{Refine}$ on this split. Essentially, we are selecting $\pi_\text{Refine}$'s best refinements on this split to use as training data for $\pi_\theta$. As such, it is an evaluation dataset for $\pi_\text{Refine}$ but a training dataset for the final model. We recognize that this is confusing, so we have changed the text to say "Table 3 shows the pass rates for $\pi_\text{Refine}$ on MBPP_Train" instead.
> > - "Table 5 is another example: the caption explicitly says "InstructRM Ensemble performs best" and that row is bolded, but two other rows are better: the naive MaxLength baseline is way better, and Prompt 2 is slightly better."
> >     - Thank you for pointing out this mis-wording -- we have corrected this table caption to simply say that InstructRM Ensemble is used throughout the paper.
> >     - The reason we still use the ensemble instead of only Prompt 2 or the naive MaxLength baseline is because we would like our technique to avoid over-tuning to a specific prompt or relying on a spurious heuristic (*i.e.* length). Even if the win rate (as judged by human raters) of the ensemble is somewhat lower than that of the MaxLength baseline, it is a common bias for human raters to pick the longer sequence of text, even if the content has not changed ([Goldberg et al.](https://arxiv.org/abs/2311.09497))
> > - "The paper says "forming dataset $\mathcal{D}=\{t\}$". This is bad notation, as it looks like a singleton set."
> >     - Thank you for this note, we have changed this notation to say "We also have a distribution of tasks $t\sim p_T$" without using it to define $\mathcal{D}$.
> > - It's bad notation to sample $t\sim p(t)$: it should be $t\sim p$ as $t$ isn't known before sampling itself.
> >     - This is true, we have revised the section to say "$t\sim p_T$."
> > - "The $\mathrm{Finetune}(\pi_\theta,\mathcal{D})$ is confusing. The notation for $\mathcal{D}$ makes it seem like it contains task inputs but not outputs, since section 2.2 clearly thinks of $t$ is being the input. But you can't fine-tune until you have the outputs."
> >     - We have removed the earlier mentions of $\mathcal{D}$ that make it seem like it consists only of the input. Now $\mathcal{D}$ can be any labelled dataset that a model can be fine-tuned on. The specific instances of $\mathcal{D}$ are made more clear in Algorithms 1 and 2.
> > - "Eqns. (6) and (7) are harder to read than just writing out the multistep sampling algorithm, and somewhat misleading since only the sampling algorithm is used, not the density (because the density is intractable). It would be better to just write the few-line algorithm showing each sampling step."
> >     - The equations for the density provide theoretical justification for the sampling procedure -- without it, it is unclear why this particular set and order of sampling steps is being undertaken. We have also revised Section 2.3 to emphasize that $q$ is only sampled from, rather than computed. The steps of the sampling algorithms are also specified in Algorithms 1 and 2.
> > - "The delta functions used are Kronecker delta functions, not Dirac delta functions."
> >     - Thank you for this correction, we have revised it.
> > - "Figure 1 was unclear when I first saw it, as I wasn't sure what the type signature of each component was. E.g., I was initially expecting there to be a model that generated feedback, but it was missing, and I wasn't sure what the type signature of $\pi_\text{Refine}$ was."
> >     - We have added further detail to the caption of Figure 1 -- are there additional changes that you think would help make this figure more clear?

---

> > > ### Author Response · Authors · 2024-01-05
> > > **Responses continued**
> > >
> > > - "The method shows the model one of the unit tests. Does it use the rest of the unit tests for rejection sampling? It's likely I just missed this, but it seems important to state explicitly so that other compares compare to this one accurately."
> > >     - In the "Dataset" portion of Section 3.2, we state the following: "Since the task descriptions are sometimes ambiguous, we include one unit test in the task description. The addition of the unit test helps to specify the input and output format of each task.  We hold out the remaining unit tests for the evaluation of our generated programs."

---

> > > ### Comment · Reviewer_zchu · 2024-01-09
> > >
> > > Thank for the detailed replies!  I think there are still some things to resolve:
> > >
> > > 1. My understanding of your baseline choices is that (1) you aspirationally don't want to need to fine-tune on human golden date, (2) because of this aspiration the ideal baseline would be zero-shot when quoting the 38% number, and (3) but actually the 38% number is against a model trained on human golden data.  This is not a reasonable way to choose a baseline: if you want zero-shot to be your baseline, you have to not use unfair data in your method.  Citing Appendix A.4 does not change this problem unless the 38% number is from Appendix A.4, and my understanding is that it isn't.  In addition, comparing GPT-<n> models which have been extensively trained on gathered human data to a pure CodeGen-Mono zero-shot is not a good comparison either: a better baseline would be one that holds the used models constant in some way, such as by using GPT-<n> for comparison purposes.  That's a bunch of work and I don't think you need to do this, but as is I do think the baseline needs to be training on the golden data.
> > >
> > > 2. Dropping the weights is fine as approximation now that it is stated explicitly, though I still think the section is going to confuse people by making them think you are doing importance sampling.  It's not likely that the weights are actually close to constant.  I won't ask for further changes here now that it's explicit.
> > >
> > > 3. Kronecker delta functions aren't centered at places, there's just one two argument $\delta_{ij}$ function.
> > >
> > > 4. I continue to think that "38% relative (and 10% absolute)" is unnecessarily confusing.  Please change it to "from X to Y" as requested.
> > >
> > > 5. Thank you for fixing the caption for Table 5.  You're still bolding the worse row, though, which should be fixed.  The rationale for using the currently bolded row is reasonable (and could even be summarized in the caption), but the best performing row should still be bolded.
> > >
> > > 6. Thank you for for the notation in improvements!
> > >
> > > 7. Yes, Figure 1 is easier to understand now.  Thank you!

---

> ### Author Response · Authors · 2024-01-21
>
> Thank you for reading our responses and getting back to us! We have made further revisions to address your comments:
>
> 1\. In table 4, we now label the method that fine-tunes on ground truth/gold MBPP data as the baseline instead, and compare to this number instead in both our abstract and in Section 3.3.
>
> 3\. We have changed the way the Kronecker delta function is described, removing the phrasing of "centered at."
>
> 4\.  We now report the increases in accuracy as x% to y% instead, in the Abstract, Introduction, and Section 3.3.
>
> 5\. We have removed the bolding in this table.
>
> Thank you again for helping us improve our paper.

---

### Review · Reviewer_MVCT · 2023-11-10

**Summary Of Contributions:**

This paper is about improving Large Language Models (LLMs) by leveraging natural language feedback that describes what is wrong with the model's output and how to fix it. The authors called this approach Imitation learning from Language Feedback (ILF). Specifically, the authors start by doing inference with $\pi_\theta$, i.e. the LLM they wish to improve on a given dataset. Next, they ask humans to annotate each output for which the model got it wrong according to some task-dependent metric. The annotation consists of 1) a natural language description of what is wrong with the output and 2) how it can be improved, and also 3) the expected improved output. With that annotated dataset in hand, they fine-tune a separate language model, $\pi_\mathrm{refine}$ such that it generate refinements given an incorrect output and natural language feedback. Equip with $\pi_{\mathrm{refine}}$, the authors can now generate refinements and use them as targets to fine-tune $\pi_\theta$. Experiments were conducted on two tasks: code generation and text summarization. Empirically, it was shown that this approach improves the LLM on those tasks compared to fine-tuning it directly on the human-written refinements.

While a lot of work has been done in the space of improving LLMs, this paper presents a novel way of leveraging natural language feedback to do so. To my knowledge, I haven't seen other work that does this and certainly as thorough. The paper is well written and structured, and all claims are backed by empirical evidence. I look forward to see more work in this direction, especially with more ILF iterations.

I recommend this paper for acceptance.

**Audience:**

Yes

**Broader Impact Concerns:**

I think a Broader Impact Statement can added to discuss the following points:
- The authors mention that the feedback is collected from crowd workers. What are the potential biases that could arise from that.
- At several occasion in the paper (e.g., Table 4), it is mentioned that language model could generate feedback and refinements (i.e. self-evaluate). What are the implication/bias of that?

**Claims And Evidence:**

Yes

**Requested Changes:**

While addressing the questions and concerns listed above would strengthen the work in my view, they are not critical for my recommendation for acceptance.

## Typos
- p.2: Missing point between "... see Section 3.3)" and "For text summarization, ..."
- p.16, Section 4.4.3. Missing Table numbers in the first paragraph.

**Strengths And Weaknesses:**

## What I like about this paper

- Well written and structured paper with many experiments that serve to demonstrate the validity of their approach.
- I appreciate the effort the authors have taken into collecting language feedback data for the two tasks and their commitment to open source it.
- The formalization of the approach and its connection to minimizing the KL divergence to the target distribution.
- My main take away is in Section 3.3, which shows that ILF performs better for the code generation task than fine-tuning directly on gold data or even human-writte programs. I believe this is an important result that can help understanding how to better self-improve LLMs in the future.

## Concerns
- I was expecting to see a note about how ILF is more data efficient than RLHF approach since $\pi_\mathrm{refine}$ can leverage pretrained language model as opposed to learning a user preference models from scratch for RLHF. I'm curious to know if the authors have any intuition on that.

- In the ablation experiment to test whether CodeGen-Mono 6.1B is good at incorporating feedback (which is needed for $\pi_\mathrm{refine}(x_1|t, x_0, f)$), the authors compared providing relevant feedback vs. unrelated feedback. I am concerned with that ablation. I could imagine the model to be performing worse because it is being distracted by the unrelated feedback, thus actually incorporating it. I think a better baseline would be to provide no feedback (or almost no feedback), e.g. "Fix the code". I don't expect the outcome to change (i.e., model + relevant feedback should be significantly better than no feedback) but that seems a more principle ablation to me.
  - If I understood correctly, the authors seem to have done it for the text summarization task (see Fig. 15 in Appendix C.9). Why not for the code generation task? Was there a particular reason for ablating with unrelated feedback?

- In section 3.3, one experiment is ablating the human annotations by using an LLM to generate feedback and refinements instead (i.e., the one the authors call InstructGPT). What was the prompt template use for this?

- In Table 4, what can explain models fine-tuned on InstructGPT 1/2-shot refinements performing worse than the zero-shot baseline? I would expect it to be at least as good as the zero-shot baseline. Is it also because of the point mentioned in "Analysis of Training Data Sources" where InstructGPT's refinements would be too much out-of-distribution compared to the data CodeGen-MONO 6.1B was trained on? If so, any ideas on how to mitigate that?

- In Section 4.1, according to equation (12), the reward function $R$ conditions on the context $c$, however I don't see it provided in the binary question used to approximate $R$: "Does this new text [$x_1$] incorporate the feedback [$f$] provided on the initial text [$x_0$]? Answer Yes or No."

---

> ### Author Response · Authors · 2024-01-05
> **Thank you for your review -- answers to your questions are below, and revisions are highlighted in red in the paper**
>
> Thank you for providing valuable feedback for our paper -- we have made multiple revisions (in red in the updated manuscript) based upon your suggestions and have provided some answers to your questions below.
> - "I was expecting to see a note about how ILF is more data efficient than RLHF approach since $\pi_\text{Refine}$ can leverage pretrained language model as opposed to learning a user preference models from scratch for RLHF. I'm curious to know if the authors have any intuition on that."
>     - We agree that this is an interesting research question -- it appears that early RLHF works such as [Stiennon et al.](https://arxiv.org/abs/2009.01325) commonly use on the order of tens of thousands of preference examples to learn a reward model. On the other hand, ILF uses <100 examples for code generation and ~5K examples for text summarization. However, it is also difficult to directly compare the data efficiency -- RLHF uses only preference comparisons, which may be easier and cheaper to collect (per example) than natural language feedback. It would be interesting to explore in future work approximately how many preference comparisons are equivalent to a single instance of natural language feedback, using the same pool of human raters and the same models + datasets. We did not have the resources to directly explore this question, but we speculate that using natural language feedback may still result in lower *total* cost, since preference comparisons cannot specify how and where the model output should be changed. It is possible that such fine-grained insights may be learned from preference data alone, but this may require learning such patterns over much larger datasets.
> - "In the ablation experiment to test whether CodeGen-Mono 6.1B is good at incorporating feedback (which is needed for $\pi_\text{Refine}(x_1|t,x_0,f)$), the authors compared providing relevant feedback vs. unrelated feedback. I am concerned with that ablation. I could imagine the model to be performing worse because it is being distracted by the unrelated feedback, thus actually incorporating it. I think a better baseline would be to provide no feedback (or almost no feedback), e.g. "Fix the code". I don't expect the outcome to change (i.e., model + relevant feedback should be significantly better than no feedback) but that seems a more principle ablation to me."
>     - We chose this comparison because it indicates whether $\pi_\text{Refine}$ is actually using the *content* of the feedback, rather than simply ignoring it and generating its own solution to the original task. We found it more difficult to interpret the comparison to having no feedback at all, since this latter setting would involve shorter prompts on average. With shorter prompts, the LLM has fewer computational steps (*i.e.* tokens) to work with, so it is difficult to disentangle the effects of variable computation versus lack of feedback.
>     - Regardless, we would have still liked to include this additional experiment for completeness, but we chose to focus our limited time/resources instead on the new experiments in Appendix A.4.
> - "In section 3.3, one experiment is ablating the human annotations by using an LLM to generate feedback and refinements instead (i.e., the one the authors call InstructGPT). What was the prompt template use for this?"
>     - Thank you for noting this minor omission -- we have revised Section 3.3 to provide more details about the prompt template. In particular, we use the same prompt templates as for $\pi_\text{Refine}$ (Appendix A.1, Fig. 7), with two slight modifications. Firstly, for the feedback prompts, we end the prompt at "Feedback:" whereas we use the entire prompt (with the previously InstructGPT-generated feedback inserted after "Feedback:") for refinement prompts. Secondly, we use $k$ in-context examples, with $k$ specified in Table 4.

---

> > ### Author Response · Authors · 2024-01-05
> >
> > - "In Table 4, what can explain models fine-tuned on InstructGPT 1/2-shot refinements performing worse than the zero-shot baseline? I would expect it to be at least as good as the zero-shot baseline. Is it also because of the point mentioned in "Analysis of Training Data Sources" where InstructGPT's refinements would be too much out-of-distribution compared to the data CodeGen-MONO 6.1B was trained on? If so, any ideas on how to mitigate that?"
> >     - Yes, we observed that InstructGPT often generated its own solutions to the problem instead of providing minimal revisions to CodeGen-Mono's solutions. (Furthermore, as demonstrated in examples in Appendix A.6, InstructGPT often gave unhelpful feedback.) When $\pi_\theta$ (CodeGen-Mono 6.1B) was then trained on InstructGPT's outputs, it would have to learn a completely different way of solving the problem. On the other hand, $\pi_\text{Refine}$ was trained to generate refinements that were as close as possible to the original model output while still incorporating the changes suggested by the feedback.
> >     - Re: mitigation -- a few generations of more recent GPT models have been released since these experiments were conducted. It is possible that using a combination of more recent (and powerful) GPT-3.5/-4 models and prompt engineering would yield better results.
> > - "In Section 4.1, according to equation (12), the reward function $R$ conditions on the context $c$, however I don't see it provided in the binary question used to approximate $R$: "Does this new text [$x_1$] incorporate the feedback [$f$] provided on the initial text [$x_0$]? Answer Yes or No.""
> >     - Apologies for the confusion, this snippet gives only an excerpt of the full prompt -- we reference the full prompt in both Section 4.3 and Appendix B.2. For clarity, we have also revised this part of Section 4.1 with a new footnote that directly links to the full prompt in Appendix B.2.
> > - "I think a Broader Impact Statement can added to discuss the following points..."
> >     - Thank you for this suggestion, we have added a Broader Impacts statement in Appendix D which discusses both how the biases of the annotators may affect the model and how multiple rounds of learning from feedback may cause bias amplification.
> > - Re: Typos
> >     - Thank you for pointing these out, we have corrected the typos.

---

> > > ### Comment · Reviewer_MVCT · 2024-01-15
> > >
> > > Thank you for the responses and clarifications you provided.

---

### Review · Reviewer_EjWp · 2023-12-22

**Summary Of Contributions:**

This paper presents a novel two-step procedure (imitation learning from language feedback, ILF) to incorporate language feedback to improve the base generative model. The feedback is first used by a refinement model whose outputs are ranked and used for finetuning the base model. ILF is instantiated on two problems: code generation and text summarization.

For code generation, CodeGen-6B model is used as the base model for both refinement and generation. The paper shows that finetuning with model-generated refinements outperforms/matches finetuning with InstructGPT generated refinements, gold answer and human written refinements on pass@k metrics.

For text summarization, GPT3-175B model is the base model and the FeedME model is the refinement model. This setup does better than FeedME, and finetuning with initial and human summaries, as judged by human evaluators. Up on using a further ranking method OPT-RM with 64 samples, ILF achieves parity with human summaries.

**Audience:**

Yes

**Claims And Evidence:**

Yes

**Requested Changes:**

Please see the weaknesses listed above with suggested changes.

**Strengths And Weaknesses:**

Strengths of the paper
---
* The paper contributes a novel finetuning technique that takes advantage of language feedback and demonstrates it on two diverse tasks.
* I liked the ablation to identify whether the feedback is actually used by the refinement model and discussion on difficulty of fixing multiple bugs pointed out in the feedback.
* The human evaluation of quality of text summarization is valuable and the authors have indicated that they will open source their code and data (including human feedback).

Weaknesses of the paper
---
Experimentation
* Code generation: Can $\pi_\theta$ and $\pi_{\text{Refine}}$ be different code generation? The datasets are small and it should be possible to use an off-the-shelf LLM capable of repair as $\pi_{\text{Refine}}$. This would substantiate the claim that "the approach ... appealing ... not model-specific".
* Text summarization: The authors use GPT3-175B as the base generative model and finetune it. Surprisingly, they do not compare the finetuned model with the base (unfinetuned) GPT3-175B model in Figure 6; they use FeedMe instead. The athours should include zero-shot and OPT-RM setups with GPT3-175B.
* The output sequences for both tasks are short (MBPP are small programs and text summarization is designed to be short). The paper does not offer any evidence that ILF can work for longer output sequences.
* ILF is presented as an iterative approach but the tasks don't use multiple iterations. The authors should either show utility of multiple iterations or mark it explicitly as future work. The claim "the approach ... multiple rounds to continuously improve ..." is an over-statement in the absence of experiments.

Presentation
* Both algorithms have mistakes: Algorithm 1, subscript k, line 2. Algorithm 2, line 11, are you proposing to create a separate $\pi_{\theta}^*$ in each iteration (then you need to select among the k-copies produced in k-iterations) or is the LHS a typo (meaning, you continue to finetune $\pi_\theta$ across iterations)?
* Insights and limitations: The paper offers interesting results but lacks insightful discussion about why the technique works. The limitations of the approach (e.g., only two tasks with small output length) etc. are not discussed adequately. The authors have sprinkled some insights (e.g., the last para in Sec 3.3 and conclusions section) and limitations here and there. I recommend that the authors create a new section to discuss some insights (if possible backed with some eval) and limitations.
* There are typos (e.g., extra ")" and missing Table ref in Sec 4.3) and citations appear without \citep in conclusions.

---

> ### Author Response · Authors · 2024-01-05
> **Thank you for your review! Answers are below, and new experiments + revisions are highlighted in red in the updated manuscript**
>
> We thank the reviewer for volunteering their time and energy towards reviewing our paper. Your feedback has helped us improve our paper, and we have included both new experiments and revisions in red in the updated manuscript.
> - "Code generation: Can $\pi_\theta$ and $\pi_\text{Refine}$ be different code generation? The datasets are small and it should be possible to use an off-the-shelf LLM capable of repair as $\pi_\text{Refine}$. This would substantiate the claim that "the approach ... appealing ... not model-specific"."
>     - Thank you for mentioning this valuable point -- we have added new experiments with different types of $\pi_\text{Refine}$ models, detailed in Appendix A.4. We compare fine-tuned CodeGen-Mono to 2-shot `gpt-3.5-turbo` and `gpt4`. ILF consistently produces improvements that significantly surpass those of the baselines and ablations, regardless of which $\pi_\text{Refine}$ one uses.
> - "Text summarization: The authors use GPT3-175B as the base generative model and finetune it. Surprisingly, they do not compare the finetuned model with the base (unfinetuned) GPT3-175B model in Figure 6; they use FeedMe instead. The athours should include zero-shot and OPT-RM setups with GPT3-175B."
>     - We compared with the performance of FeedME (`text-davinci-001`) instead of GPT3-175B (`davinci`) for a number of reasons -- (1) the original summaries that we sought to provide feedback for were sourced from FeedME; (2) at the time, OpenAI did not provide fine-tuning access to FeedME so we could not simply fine-tune FeedME instead; and (3) FeedME is considered the more advanced model, so this is a stronger baseline to compare to. Since we evaluated win rates with human evaluations, it was also too costly to compare the final results to both FeedME and GPT3-175B. Given this constraint, we believe that comparing to the stronger of the two models (and the source of the original summaries) provides a more meaningful comparison.
> - "The output sequences for both tasks are short (MBPP are small programs and text summarization is designed to be short). The paper does not offer any evidence that ILF can work for longer output sequences."
>     - It would be interesting to explore the effectiveness of ILF for longer output sequence tasks -- however, even most long-context models take long contexts while outputting shorter outputs (*e.g.* `gpt-4-1106-preview` takes 128K tokens but only outputs 4K tokens - [source](https://platform.openai.com/docs/models/gpt-4-and-gpt-4-turbo)). We plan to explore this aspect of ILF in future work, once there exist more models capable of returning longer output sequences.
> - "ILF is presented as an iterative approach but the tasks don't use multiple iterations. The authors should either show utility of multiple iterations or mark it explicitly as future work. The claim "the approach ... multiple rounds to continuously improve ..." is an over-statement in the absence of experiments."
>     - We provided limited results on applying ILF over multiple rounds of text summarization in Appendix C.8.2. In short, applying ILF iteratively over multiple rounds can improve over training on the same number of refinements in a single round. However, doing too many rounds of continuous fine-tuning can also result in some degree of forgetting. Our experiments suggest that this forgetting can be remedied by instead fine-tuning just once over the combined refinements gathered over multiple rounds of ILF. However, these are limited experiments that require a more thorough and elaborate analysis in future work. For now, we focus on the more robust and promising results we have provided from a single round of ILF.
> - "Both algorithms have mistakes: Algorithm 1, subscript k, line 2. Algorithm 2, line 11, are you proposing to create a separate $\pi_\theta^*$ in each iteration (then you need to select among the k-copies produced in k-iterations) or is the LHS a typo (meaning, you continue to finetune $\pi_\theta$ across iterations)?"
>     - Thank you for pointing out the minor typo in Algorithm 1 -- we have removed the $k$ subscript. Only a single $\pi_\theta$ model is fine-tuned in each round of ILF. Since our experiments focus only on a single round of ILF, our algorithms also describe only a single round. Once $\pi_\theta$ has been fine-tuned, we refer to the fine-tuned model as $\pi_{\theta^*}$.

---

> > ### Author Response · Authors · 2024-01-05
> > **Responses continued**
> >
> > - "Insights and limitations: The paper offers interesting results but lacks insightful discussion about why the technique works. The limitations of the approach (e.g., only two tasks with small output length) etc. are not discussed adequately. The authors have sprinkled some insights (e.g., the last para in Sec 3.3 and conclusions section) and limitations here and there. I recommend that the authors create a new section to discuss some insights (if possible backed with some eval) and limitations."
> >     - We have revised the Conclusion to include these limitations (combined with suggestions for future work).
> >     - We also agree that it is useful to discuss why the technique may work -- we included multiple sections and analyses that discuss this topic.  For instance, Section 2 provides theoretical justification for the effectiveness of ILF. Figure 3 and the subsection labelled "Analysis of Training Data Sources" in Section 3.3 discusses and analyzes why $\pi_\text{Refine}$-generated code refinements may be easier for the base model to learn from than the other datasets. We also discuss in Appendix A.6 how ILF may be less effective when the feedback must address many points, since $\pi_\text{Refine}$ is less effective the more bugs there are in the original output. Section 3.4 discusses how ILF may be more effective than learning from model-generated feedback due to the higher quality of human feedback (accompanied by analyses comparing human versus InstructGPT-generated feedback in Appendices A.5 and A.7). Section 4.4 and Appendix C.8 discuss how the final fine-tuned model ($\pi_{\theta^*}$) exhibits lower loss on the validation set of refinements than the human-written summaries, suggesting that $\pi_\theta$ learns more effectively on the former dataset. The subsection labelled "Combining ILF With Learning From Binary Feedback" in Section 4.4.3 analyzes how natural language feedback may provide additional learning signal that isn't easily learned from binary feedback alone. Appendix C.8.2 uses experimental evidence from multiple rounds of ILF to demonstrate how training on the data gathered over multiple iterations continues to boost accuracy more than training on off-policy data.
> > - "There are typos (e.g., extra ")" and missing Table ref in Sec 4.3) and citations appear without \citep in conclusions."
> >     - We unfortunately could not locate the missing `\citep` in the conclusions, but we have fixed the missing table reference in Section 4.3. Thank you for mentioning these.

---

> > ### Comment · Reviewer_EjWp · 2024-01-17
> >
> > Thank you for your detailed response and the revised version! It addresses several of my comments.
> >
> > There are a few points that I still find unresolved:
> > 1. Text summarization - Thank you for clarifying why you chose to finetune GPT3 instead of FeedME. However, I think it is fair to ask for comparison to zero-shot GPT3 because that's the model that has been finetuned. I find that to be an obvious gap.
> >
> > 2. Output sequences are short - I find the response tangential. My comment is that MBPP generations and text summarizations are quite short - few tens to a hundred-two hundred tokens maybe. I am neither referring to long-context models nor very long generations (exceeding 4K tokens), but generations say upwards of 500 tokens.
> >
> > 3. Algorithm 2, $\pi_{\theta*}$ - Perhaps my point was not clear. Algorithm 2 is an iterative algorithm with the outer loop doing $K$ iterations. It is fine if you are using $K=1$ in the experiments but algorithmically, $\pi_{\theta*}$ is a variable that you are assigning to in every iteration ($\pi_{\theta}$ finetuned with $D_k$). So $\pi_{\theta*}$ will have the result of the last $k = K$th iteration when the algorithm terminates ($\pi_{\theta}$ finetuned with $D_k$ for $k=K$). The inner loop uses the initiate weights $\pi_{\theta}$ and recomputes $D_k$ in each loop. Then what is the point of running the previous $k-1$ iterations?

---

### Decision · Action_Editor_L5QT · 2024-02-08

**Recommendation:** Accept as is

**Comment:**

The paper proposes Imitation Learning from Language Feedback (ILF) which involves enhancing a language model (LLM). First, they performing inference on a dataset and identify incorrect outputs based on a task-specific metric. Then, humans annotate the outputs with a description of the error, suggestions for improvement, and the expected correct output. This annotated dataset is used to fine-tune a separate model to generate refinements for incorrect outputs given feedback. These refinements are then used as targets for further fine-tuning the original LLM.

All reviewers agree that this paper makes interesting contributions, was very thorough in the experimental framework. One of the main contributions found by a reviewer is that ILF performs better for the code generation task than fine-tuning directly on gold data or even human-written programs. A particular interesting ablation was to identify whether the feedback was actually used by the refinement model and an analysis on the difficulty of fixing multiple bugs pointed out in the feedback. One of the weaknesses pointed out by a reviewer is that the length of the outputs in the tasks considered is rather small, this particular point might be highlighted in the camera ready version.

Overall this is a strong and sound paper. The release of the human feedback dataset also makes a strong contribution to the field.

**Audience:**

Yes, especially audience interested in how to refine LLMs based on natural language feedback. The authors commit to open source language feedback data for the two tasks, which will be useful to a variety of research streams.

**Claims And Evidence:**

Claims are accurate and supported by evidence.